# CTP promotes efficient ParB-dependent DNA condensation by facilitating one-dimensional diffusion from *parS*

Francisco de Asis Balaguer[1], Clara Aicart-Ramos[1], Gemma LM Fisher[2†], Sara de Bragança[1], Eva M Martin-Cuevas[1], Cesar L Pastrana[1‡], Mark Simon Dillingham[2]*, Fernando Moreno-Herrero[1]*

[1]Department of Macromolecular Structures, Centro Nacional de Biotecnología, Consejo Superior de Investigaciones Científicas, Madrid, Spain; [2]DNA:Protein Interactions Unit, School of Biochemistry, Biomedical Sciences Building, University of Bristol, University Walk, Bristol, United Kingdom

**Abstract** Faithful segregation of bacterial chromosomes relies on the ParABS partitioning system and the SMC complex. In this work, we used single-molecule techniques to investigate the role of cytidine triphosphate (CTP) binding and hydrolysis in the critical interaction between centromere-like *parS* DNA sequences and the ParB CTPase. Using a combined optical tweezers confocal microscope, we observe the specific interaction of ParB with *parS* directly. Binding around *parS* is enhanced by the presence of CTP or the non-hydrolysable analogue CTPγS. However, ParB proteins are also detected at a lower density in distal non-specific DNA. This requires the presence of a *parS* loading site and is prevented by protein roadblocks, consistent with one-dimensional diffusion by a sliding clamp. ParB diffusion on non-specific DNA is corroborated by direct visualization and quantification of movement of individual quantum dot labelled ParB. Magnetic tweezers experiments show that the spreading activity, which has an absolute requirement for CTP binding but not hydrolysis, results in the condensation of *parS*-containing DNA molecules at low nanomolar protein concentrations.

*For correspondence:
mark.dillingham@bristol.ac.uk
(MSD);
fernando.moreno@cnb.csic.es
(FM-H)

Present address: †Cell cycle group, MRC London Institute of Medical Sciences (LMS), London, United Kingdom; ‡Physics of Complex Biosystems, Physics Department, Technical University of Munich, Munich, Germany

Competing interests: The authors declare that no competing interests exist.

## Introduction

In bacterial cells, the separation of sister chromosomes is performed by the ParABS system and the SMC complex (*Britton et al., 1998*; *Ireton et al., 1994*; *Jensen and Shapiro, 1999*; *Marbouty et al., 2015*; *Mohl and Gober, 1997*; *Song and Loparo, 2015*; *Wang et al., 2014*). The ParABS system consists of the ATPase protein ParA, the DNA-binding protein ParB, and a centromere-like palindromic DNA sequence named *parS* (*Funnell, 2016*; *Lin and Grossman, 1998*). *In vivo* imaging experiments in *Bacillus subtilis* showed that multiple ParB proteins co-localize with SMC complexes at a given *parS* site forming distinctive clusters in the cell (*Gruber and Errington, 2009*; *Minnen et al., 2011*; *Sullivan et al., 2009*). Notably, chromatin immuno-precipitation (ChIP) experiments indicate that ParB covers regions of up to 18 kilobase pairs (kbp) of DNA surrounding *parS* (*Breier and Grossman, 2007*; *Graham et al., 2014*; *Minnen et al., 2016*; *Murray et al., 2006*; *Rodionov et al., 1999*), a phenomenon named *spreading*. Originally, this spreading was interpreted as the formation of a nucleoprotein filament extending from a *parS* nucleation site (*Murray et al., 2006*; *Rodionov et al., 1999*). However, it later became clear that ParB foci contained far too few proteins to coat tens of kbp-long DNA segments (*Graham et al., 2014*). Instead, we and others have shown that ParB can self-associate to form networks which include specific binding to *parS* sequences but also non-specific binding to distal DNA segments. Overall, this results in the condensation and bridging of DNA at low forces (below 1 pN) (*Fisher et al., 2017*; *Graham et al., 2014*;

*Madariaga-Marcos et al., 2019*; *Madariaga-Marcos et al., 2018*; *Taylor et al., 2015*), and could explain how distant regions of DNA are bound by limited numbers of ParB proteins as shown in ChIP experiments. However, DNA condensation has only been observed *in vitro* at high ParB concentrations, in the low micromolar range. Moreover, and unexpectedly, *parS* sequences did not affect DNA condensation under these conditions (*Taylor et al., 2015*). Therefore, the mechanism of ParB spreading and condensation, and in particular the molecular basis for the specific localization around *parS*, has remained unclear despite extensive investigation *in vivo*, *in vitro*, and *in silico* (*Broedersz et al., 2014*; *Guilhas et al., 2020*; *Sanchez et al., 2015*; *Walter et al., 2020*).

ParB proteins comprise three distinct domains (*Figure 1A and B*). The N-terminal domain (NTD) binds ParA (*Bouet and Funnell, 1999*; *Davis et al., 1992*; *Radnedge et al., 1998*; *Vecchiarelli et al., 2010*) and was recently appreciated to contain a CTP-binding pocket that serves as a CTP-dependent dimerization interface (*Osorio-Valeriano et al., 2019*; *Soh et al., 2019*). Mutation R80A in the CTP-binding pocket has been shown to impair nucleoid segregation and ParB spreading (*Autret et al., 2001*; *Breier and Grossman, 2007*). A central DNA-binding domain (CDBD) binds specifically to the palindromic *parS* sequence and may also facilitate dimerization (*Leonard et al., 2004*; *Schumacher and Funnell, 2005*). In this central region, the mutation R149G within a helix-turn-helix motif impedes *parS* binding (*Autret et al., 2001*; *Fisher et al., 2017*; *Gruber and Errington, 2009*). Finally, the C-terminal domain (CTD) forms dimers with a lysine-rich surface that binds to DNA non-specifically (*Fisher et al., 2017*). Interactions between CTDs are essential for condensation *in vitro* and for the formation of ParB foci *in vivo* (*Fisher et al., 2017*). Based upon these results, we proposed a model for DNA condensation dependent on ParB CTD dimerization and non-specific DNA (nsDNA) binding (*Fisher et al., 2017*). Further work using lateral pulling of long DNA molecules in a magnetic tweezers (MT) setup combined with total internal reflection fluorescence microscopy (TIRFM) demonstrated that ParB networks are highly dynamic and display a continual exchange of protein-protein and protein-DNA interfaces (*Madariaga-Marcos et al., 2019*).

Recently, two laboratories demonstrated independent that *B. subtilis* ParB and the *Myxococcus xanthus* ParB hydrolyse cytidine triphosphate (CTP) to cytidine diphosphate (CDP) (*Osorio-Valeriano et al., 2019*; *Soh et al., 2019*) and require CTP for partition complex formation *in vivo* (*Osorio-Valeriano et al., 2019*). Importantly, *parS* DNA stimulates the binding and hydrolysis of CTP (*Osorio-Valeriano et al., 2019*; *Soh et al., 2019*), CTP has been shown to be necessary for ParB spreading *in vitro* (*Jalal et al., 2020*), and a model has been proposed in which centromeres assemble via the loading of ParB-DNA sliding clamps at *parS* sequences (*Jalal et al., 2020*; *Soh et al., 2019*). These observations fundamentally change our understanding of how ParB can become engaged with nsDNA surrounding *parS*, but the significance of CTP-dependent spreading in the formation of the ParB networks that cause DNA bridging and condensation has not been addressed.

Here, we investigated the role of CTP binding and hydrolysis in the binding of ParB to *parS* sequences and nsDNA at the single-molecule level. We present the first visualization of the specific binding of ParB to *parS*. ParB proteins were also detected at nsDNA far from *parS*, albeit at a much lower density. Importantly, this only occurred in *parS*-containing DNA, suggesting spreading from *parS* sites. The placement of tight-binding protein roadblocks on DNA constrained the spreading, suggesting it arises from one-dimensional movement along the contour of DNA, consistent with a sliding clamp model. CTP binding mediated by a $Mg^{2+}$ cofactor, but not CTP hydrolysis, was critical for spreading from *parS* to non-*parS* sites which triggered DNA condensation at nanomolar protein concentration. We propose a model where ParB-CTP-$Mg^{2+}$ loads to *parS*, diffuses to non-*parS* sites, and then self-associates, resulting in the condensation of kbp-long DNA molecules.

## Results

### Direct visualization of the specific binding of ParB to *parS* sequences

In our previous work, performed in the absence of CTP, we used a combination of TIRFM and DNA stretching (by either flow or MTs) to visualize binding of ParB to long DNA molecules (*Madariaga-Marcos et al., 2019*). Although specific ParB-*parS* interactions have been detected in electrophoretic mobility shift assays (EMSA) (*Taylor et al., 2015*), they have not yet been observed directly. Instead, regardless of the presence or absence of *parS* sequences, we and others observe a uniform

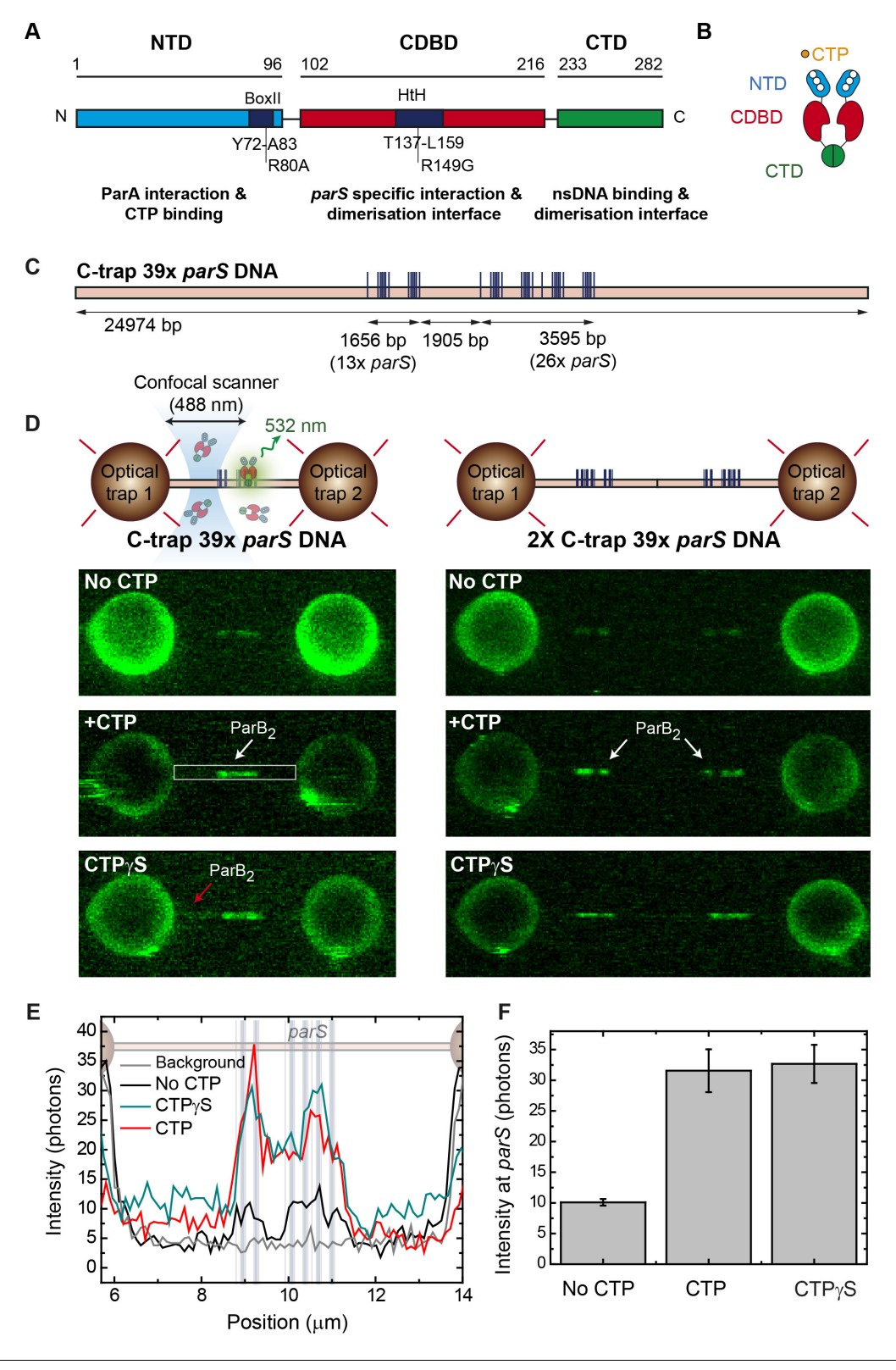

**Figure 1.** Direct visualization of ParB specific binding to *parS* sites. (A) Domains and functional motifs of ParB as reported previously (*Bartosik et al., 2004*; *Kusiak et al., 2011*). Mutations R80A, defective for cytidine triphosphate (CTP) binding, and R149G, defective for *parS* binding, are indicated. (B) ParB dimer cartoon showing dimerization through the central and C-terminal domains. The nucleotide binding site at the N-terminal domain

*Figure 1 continued on next page*

*Figure 1 continued*

(NTD) is also indicated. (**C**) Schematic representation of the single-length 39× *parS* DNA used for C-trap experiments. The DNA contains 39 *parS* sequences distributed in six groups forming two clusters separated by 1905 bp (39× *parS* DNA). The positions of the *parS* sites in the DNA cartoon are represented to scale. (**D**) Schematic of the C-trap experiment where single and tandem (double-length) tethers are immobilized between two beads and scanned with a confocal microscope using 488 nm illumination (upper part). Representative confocal images of the experiment under no CTP, 2 mM CTP, or 2 mM CTPγS conditions (lower part) and 20 nM ParB$_2^{AF488}$. Dark to bright regions correspond to a scale of 0–30 photon counts for single-length tethers and 0–50 counts for tandem tethers. (**E**) Representative profiles (500 nm width) of the fluorescence intensity along the DNA axis of the confocal images depicted in **D** (only single-length tether data). Positions of the *parS* sequences are included to scale in the background. Brighter regions between the beads correlate with the position of the *parS* clusters. ParB proteins are also observed outside the *parS* region (red arrow) and in general the fluorescence intensity outside the *parS* region is always above the background and larger in CTPγS compared to CTP experiments. (**F**) Quantification of fluorescence intensity at the *parS*-containing region under no CTP, CTP, and CTPγS conditions.

The online version of this article includes the following source data and figure supplement(s) for figure 1:

**Source data 1.** Source data file for *Figure 1*.
**Figure supplement 1.** C-trap layout and nucleotide triphosphate (NTP) hydrolysis experiments.
**Figure supplement 1—source data 1.** Source data file for *Figure 1—figure supplement 1*.
**Figure supplement 2.** Fabrication of small DNA plasmids.
**Figure supplement 3.** Fabrication of large DNA plasmids.
**Figure supplement 4.** Kymographs of ParB bound along *parS* DNA.
**Figure supplement 4—source data 1.** Source data file for *Figure 1—figure supplement 4*.

non-specific coating of DNA with ParB accompanied by rapid DNA bridging and condensation (*Graham et al., 2014*; *Madariaga-Marcos et al., 2019*).

In this study, we investigated the effect of CTP on the specific and non-specific binding of ParB to *parS* using a fluorophore-conjugated ParB$^{AF488}$, which retains specific and nsDNA-binding activity (*Madariaga-Marcos et al., 2019*), and an experimental setup that combines confocal fluorescence microscopy with dual optical tweezers (C-trap, Lumicks) (*Candelli et al., 2011*; *Newton et al., 2019*) (see Materials and methods, *Figure 1* and *Figure 1—figure supplement 1*). In this approach, DNA molecules are perpendicular to the optical axis of the microscope providing a homogeneous illumination along the molecule and confocal imaging provides a very high signal-to-noise ratio (albeit with a limited spatial resolution of around 250 nm). Additionally, the force applied to DNA is better defined and more uniform than in flow-stretch experiments.

Single DNA molecules were immobilized between two polystyrene beads and extended to almost their contour length by a force of ~20 pN. To amplify the potential signal from specific binding to *parS* sites, we used a DNA substrate that contains 39 copies of the partially degenerate *parS* sequence (5′-TGTTCCACGTGAAACA) (*Breier and Grossman, 2007*; *Taylor et al., 2015*) arranged in two clusters (*Figure 1C* and *Figure 1—figure supplement 2* and *Figure 1—figure supplement 3*). Then, we incubated the DNA with 20 nM ParB$^{AF488}$ and took confocal images of the region of interest, including the beads as a reference, in the presence and absence of CTP-Mg$^{2+}$ (*Figure 1D*). The images clearly showed two bright regions, one larger than the other, corresponding to the two *parS* clusters separated by 1905 bp. Note that, due to the design of the molecule (see Materials and methods), double-length substrates can also be generated and trapped between the beads. In this case, the number of bright clusters were doubled, as expected (*Figure 1D*). Fluorescence intensity profiles of ParB correlated with the position of the *parS* clusters (*Figure 1E*). Importantly, ParB binding to *parS* did not require CTP in agreement with previous EMSA (*Taylor et al., 2015*) and bio-layer interferometric analysis (*Jalal et al., 2020*). However, the presence of CTP enhanced the fluorescence intensity within the *parS* clusters by three- to four-fold compared to in the absence of CTP (*Figure 1F*). Previously, we showed that the ParB binding equilibrium is established rapidly (within tens of seconds; *Madariaga-Marcos et al., 2019*) compared to our incubation time before confocal imaging. Therefore, these images represent the steady-state occupancy of ParB on DNA and the higher fluorescence intensity measured in the CTP case reflects a greater number of ParB molecules bound at or around the *parS* sequences compared to the no CTP condition. It is formally possible that the higher fluorescence measured in the presence of CTP could reflect a

fluorescence enhancement effect, but this is unlikely since the ParB labelling site (S68C) is on a surface-exposed loop that is distant from the buried CTP molecules (*Soh et al., 2019*). In some images, a faint fluorescence signal was also observed outside the *parS* region (see red arrow in *Figure 1D*) in the CTP and CTPγS conditions and we will return to this point later. Control experiments with non-*parS* DNA did not show any protein binding at this ParB concentration (see below). This is the first direct visualization of ParB association specifically and precisely at *parS* sequences, and shows that this interaction is mediated by CTP binding in agreement with recently published works (*Jalal et al., 2020*; *Osorio-Valeriano et al., 2019*; *Soh et al., 2019*).

Next, we investigated the dynamics of the ParB-DNA interaction by taking kymographs with CTP/CTPγS or in the absence of nucleotide (see Materials and methods). The fluorescence intensity at the *parS* sites decayed with time but ParB remained visible for 30 s in both CTP/CTPγS conditions (*Figure 1—figure supplement 4A and B*). This helped to reveal the positioning of ParB relative to the *parS* sites throughout the 30 s kymograph (*Figure 1—figure supplement 4C*). The 39× *parS* DNA substrate includes two sequence clusters separated by 1905 bp, the smaller of which contains two groups of closely spaced *parS* sequences, and the larger of which contains four such groups (*Figure 1—figure supplement 4C*). The 30 s average intensity profile clearly distinguished six foci corresponding to groups of ParB molecules precisely at their expected positions. The gradual decay of the fluorescence over tens of seconds (*Figure 1—figure supplement 4D*) indicates that the photobleaching kinetics are faster than the rates of ParB binding and unbinding. If the opposite were true, then efficient protein turnover would result in a constant fluorescence level as was observed in our previous experiments performed at much higher ParB concentration (*Madariaga-Marcos et al., 2019*).

## ParB spreading from *parS* sites occurs by sliding and requires CTP binding but not hydrolysis

With the aim of exploring ParB spreading from *parS* sites to distal DNA sites, we performed experiments in which we incubated the ParB protein with the DNA for 2 min before illuminating the sample (*Figure 2A*). By doing this, we prevented photobleaching of the proteins in the process of loading and spreading. We then compared the fluorescence intensity profiles of the first images obtained after incubation under different experimental conditions (*Figure 2B*). Importantly, ParB proteins were now more clearly identified outside of the *parS* region (compared to *Figure 1D*), but only under CTP or CTPγS conditions (*Figure 2B*). Indeed, ParB proteins were sparsely distributed along the entire non-specific region (i.e., they did not only accumulate at or near the *parS* cluster) (red arrow, *Figure 2B* and *Figure 2C*). The CDP case was also studied (data not shown) and produced intensities very similar to the no nucleotide case (*Figure 2C*). This is consistent with the negligible affinity of this nucleotide for ParB reported by *Osorio-Valeriano et al., 2019*. The intensity in this non-*parS* region decayed within 1–2 s, an apparently shorter timescale than in the *parS* regions (*Animation 1*), re-enforcing the idea that protein turnover is slow compared to photobleaching. Crucially, a control experiment using a non-*parS* DNA showed no protein bound at all in the presence of CTP (*Figure 2B and C*), supporting the notion that proteins located outside *parS* reached that location through the *parS* entry site.

Previous in vitro and in vivo experiments have shown that ParB spreading is hindered by DNA-binding protein 'roadblocks' engineered close to *parS* (*Murray et al., 2006*; *Rodionov and Yarmolinsky, 2004*; *Soh et al., 2019*). Therefore, to investigate the mechanism by which ParB spreads to non-specific sites, we fabricated a 17 kbp DNA molecule that contains the same 39× *parS* cluster flanked by two groups of 5× EcoRI sites (*Figure 2D*). These sequences would act as roadblocks after binding of EcoRI[E111G], a catalytically inactive variant of the EcoRI restriction enzyme which has been used as a model protein roadblock (*Figure 2E*; *Brüning et al., 2018*; *King et al., 1989*). Note that, if movement of ParB occurs along the contour of DNA (i.e., by sliding from *parS*), then we expect ~5 and ~1 kbp DNA segments of the molecule to remain free of ParB proteins. Additionally, the molecule contains a ~3.6 kbp non-*parS* area between the last *parS* sequence of the 39× *parS* cluster and one of the 5× EcoRI sites which we would expect to become populated with ParB via a sliding mechanism (*Figure 2D*). The imaging experiments were performed as described in *Figure 2A* but included an additional incubation step with 100 nM EcoRI[E111G], prior to incubation with ParB[AF488]. The confocal laser was turned on after the incubations with EcoRI[E111G] and ParB[AF488], and confocal images were obtained on a tandem EcoRI 39× *parS* DNA (*Figure 2F*). As expected, brighter regions

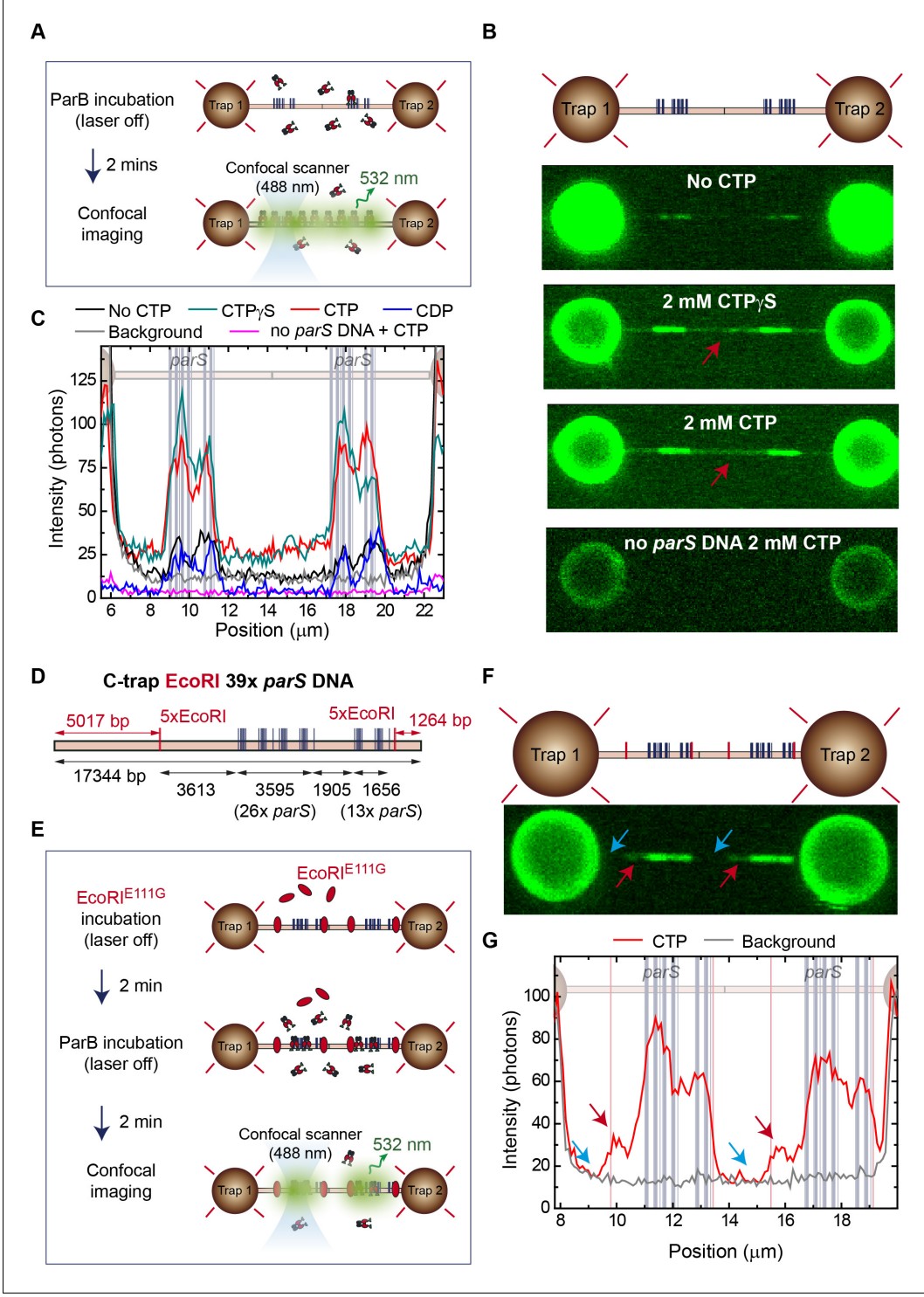

**Figure 2.** Cytidine triphosphate (CTP) binding promotes ParB spreading from *parS*. (**A**) Cartoon of the experiment. First, a tandem 39× *parS* DNA molecule is incubated with 20 nM ParB₂ and 2 mM CTP-Mg²⁺. Then, following a 2 min incubation, the confocal laser is turned on and confocal images are taken. (**B**) Representative confocal images taken after 2 min ParB incubation in the dark using tandem 39× *parS* DNA under no CTP, CTP, or CTPγS conditions, as well as *parS*-free DNA (lambda DNA) and 2 mM CTP-Mg²⁺. ParB appears in non-*parS* regions only when using *parS* DNA and under CTP or CTPγS conditions (red arrows). Dark to bright regions correspond to a scale of 0–50 photon counts for *parS* DNA tethers and 0–25 counts for lambda DNA. (**C**) Corresponding average profiles (500 nm width) of the fluorescence intensity taken along the DNA axis of the confocal images, including

*Figure 2 continued on next page*

*Figure 2 continued*

the cytidine diphosphate (CDP) case (scan not shown). Positions of the *parS* sequences are included to scale in the background. (D) Schematic representation of the single-length EcoRI 39× *parS* DNA used for C-trap roadblock experiments. The DNA contains 39 *parS* sequences arranged as in *Figure 1C*, but also includes two groups of 5× EcoRI sites flanking the *parS* region. Note that one of the 5× EcoRI groups is located 3613 bp away from the last *parS* sequence, potentially allowing spreading from the *parS* region. The positions of the *parS* sites in the DNA cartoon are represented to scale. (E) Cartoon of the roadblock experiment designed to limit ParB spreading using the EcoRI[E111G] mutant as a roadblock. The experiment is identical to that described in A, but first includes a 2 min pre-incubation with 100 nM EcoRI[E111G], which is capable of DNA binding to EcoRI sites but unable to cleave the DNA, thus acting as a roadblock. (F) Confocal image showing limited spreading due to EcoRI[E111G] blocking in tandem EcoRI 39× *parS* DNA. Brighter regions correspond to *parS* binding and the two dimmed regions correspond to limited spreading up to the EcoRI sites (red arrows). Regions inaccessible to ParB spreading are indicated with blue arrows. (G) Corresponding average profile (500 nm width) of the fluorescence intensity taken along the DNA axis of the confocal image. Positions of the *parS* sequences and EcoRI sites are included to scale in the background. Red arrows indicate the limited spreading of ParB up to EcoRI sites. Blue arrows indicate inaccessible regions to ParB.

The online version of this article includes the following source data and figure supplement(s) for figure 2:

**Source data 1.** Source data file for *Figure 2*.

**Figure supplement 1.** Cytidine triphosphate (CTP) binding promotes ParB spreading from *parS* in 7× *parS* substrates.

---

correlated very well with the *parS* clusters. Importantly, a faint region also appeared flanking the *parS* cluster and bordered by the EcoRI[E111G] roadblocks, which appeared to have constrained ParB spreading (see red arrows, *Figure 2F*). Indeed, fluorescence profiles showed high intensity associated with the *parS* region, lower intensity signals produced by ParB spreading from *parS* (see red arrows, *Figure 2F* and *Figure 2G*), and no signal associated with regions protected by EcoRI sites (see blue arrows, *Figure 2F* and *Figure 2G*). Subsequent confocal images reflected the photobleaching of this region in contrast with the brighter *parS* area, confirming the low exchange of proteins outside *parS* (*Animation 2*). Control experiments omitting the EcoRI[E111G] incubation step reproduced our previous result with a uniform intensity profile along the whole DNA molecule (data not shown). Moreover, in order to exclude a potential confining effect due the high number of *parS* used in our substrates, we repeated these experiments using a DNA containing 7 copies of *parS* (C-trap EcoRI 7× *parS* substrate, *Figure 2—figure supplement 1A*). Again, we observed enhanced binding at the *parS* sequences and blocking of ParB sliding by EcoRI[E111G] (*Figure 2—figure supplement 1B–D*). Altogether, these experiments show that CTP binding promotes movement of ParB over kbp distances away from *parS* sites to non-specific regions of DNA. The fact that this movement can be constrained by protein roadblocks suggests that it occurs by sliding from *parS*.

## Direct visualization of ParB diffusion from *parS* sequences

With the aim of capturing the movement of individual ParB dimers along the DNA molecules, we generated quantum dot (QD) labelled ParB (ParB[QD], see Materials and methods). QD labelling increased the fluorescence intensity of a single ParB protein and allowed much longer visualization times due to the absence of photobleaching. We used C-trap substrates including 7 or 2 copies of

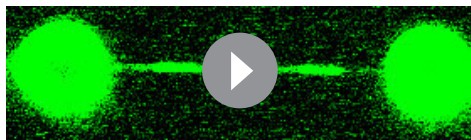

**Animation 1.** ParB binding to *parS* and spreading to non-*parS* region. Video built from individual C-trap scans showing ParB binding to *parS*, spreading to non-*parS* DNA, and photobleaching of the fluorescence dye.

https://elifesciences.org/articles/67554#video1

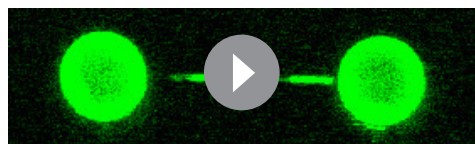

**Animation 2.** ParB spreading is limited by protein roadblocks. Video built from individual C-trap scans showing ParB binding to *parS*, and spreading to non-*parS* DNA up to the region confined by EcoRI[E111G], and photobleaching of the fluorescence dye.

https://elifesciences.org/articles/67554#video2

*parS* and recorded kymographs of several tens of seconds (*Figure 3A and B*, respectively). A clear and stable signal was observed at the *parS* region consistent with ParB binding to *parS*, as expected from previous experiments. However, ParB proteins also left the *parS* site, randomly explored the whole non-*parS* region and, very often, detached from nsDNA. We then measured the position of ParB along the DNA molecule (i.e., outside *parS*) for a given time (*Figure 3C*) and determined the mean square displacement (MSD) for a time interval ($\Delta t$) (*Figure 3D*) to obtain the diffusion constant (*D*) of ParB (see Materials and methods) (*Gorman and Greene, 2008*; *Heller et al., 2014*). The diffusion constant of ParB was $0.41 \pm 0.02$ $\mu m^2$ $s^{-1}$ (mean±SEM, $n=177$) or $3.5 \pm 0.2 \times 10^6$ $bp^2$ $s^{-1}$, assuming a rise per base pair of 0.34 nm. In contrast, ParB proteins bound to *parS* remained mostly immobile (*Figure 3F*). These data are consistent with previous single-molecule experiments performed in the absence of CTP (*Graham et al., 2014*) and with the recently calculated ParB diffusion constant of 0.7 $\mu m^2$ $s^{-1}$ using super-resolution microscopy (*Guilhas et al., 2020*). In addition, Guilhas et al. also reported a much lower value of 0.05 $\mu m^2$ $s^{-1}$ for ParB clustered in *parS* nucleoprotein condensates, which likely reflects our observation of immobile ParB proteins at *parS* (*Figure 3F*). Experiments using CTPγS produced very similar results, with clear diffusive behaviour in non-*parS* regions and stable binding of ParB at *parS* (*Figure 3—figure supplement 1*). The ParB diffusion constant obtained with CTPγS was $0.38 \pm 0.02$ $\mu m^2$ $s^{-1}$ (mean±SEM, $n=185$). These experiments directly demonstrate diffusion of individual ParB proteins along nsDNA and provide a rationale for the fluorescence intensities captured in C-trap scans on *parS* and non-*parS* regions (*Figure 1* and *Figure 2*).

## CTP binding dramatically enhances the *parS* sequence specificity of ParB-dependent DNA condensation

We have previously shown that ParB condenses DNA and that the CTD plays an important role in this function (*Fisher et al., 2017*; *Taylor et al., 2015*). However, condensation occurred at micromolar protein concentration and was not apparently specific to *parS*-containing DNA. Now, we aimed to revisit these experiments in the light of the discovery of CTP as an important mediator of ParB-DNA interactions (*Osorio-Valeriano et al., 2019*; *Soh et al., 2019*). Note that the optical trap experiments described above were necessarily performed at forces which are non-permissive for condensation, to keep the DNA extended for optimal fluorescence visualization. We therefore switched to an MT setup which is more appropriate for low-force experiments (*Taylor et al., 2015*; *Figure 4A*). Single DNA molecules containing a set of 13× *parS* sequences were immobilized between a glass surface and super-paramagnetic beads (*Figure 4B*). A pair of magnets were then employed to stretch the DNA and apply forces in the 0.1–5 pN range. The DNA was incubated with ParB at the higher force level for 2 min, and then the force was lowered to 0.33 pN, which is permissive for DNA condensation. The extension of the tether was monitored in real time leading to condensation time course plots.

DNAs containing *parS* rapidly condensed in assays with 50 nM ParB$_2$, CTP, and Mg$^{2+}$ (*Figure 4C*). In fact, DNA condensation was observed at even lower concentrations of 5–10 nM, but the rate was markedly slower (*Figure 4—figure supplement 1A*). In the presence of CTP-Mg$^{2+}$, DNA was condensed by ParB at forces of up to 1 pN using only 10 nM protein (*Figure 4D*). This maximum condensation force was similar to that described in non-*parS* DNA using micromolar ParB concentrations (*Taylor et al., 2015*), suggesting a similar mechanism of condensation. Experiments using ATP, UTP, or GTP did not produce any DNA condensation, confirming the specificity of ParB for CTP and linking CTP binding to condensation (*Figure 4E*).

The recent crystal structure of the *M. xanthus* ParB-like protein PadC showed that Mg$^{2+}$ is a cofactor of CTP at the CTP-binding site (*Osorio-Valeriano et al., 2019*). Therefore, we explored the role of Mg$^{2+}$ in the ParB condensation function. ParB did not induce any condensation in a buffer containing 1 mM EDTA and 200 nM ParB$_2$ (*Figure 4F*). Mutation of the R80 residue of ParB to alanine leads to loss of function in *B. subtilis* and impairs subcellular localization of ParB (*Autret et al., 2001*; *Graham et al., 2014*). Additionally, CTP binding is also abolished in *Bs*ParB$^{R80A}$ (*Soh et al., 2019*). We therefore ask if this mutation might affect the condensation function of ParB. Indeed, MT experiments showed no DNA condensation by ParB$^{R80A}$ under CTP-Mg$^{2+}$ conditions (*Figure 4G*) supporting the idea that CTP binding is required for condensation at nanomolar ParB concentrations. In the absence of CTP, ParB was unable to condense *parS*-containing DNA unless its concentration was raised to the micromolar range, as reported previously (*Figure 4—figure supplement 1B and C*; *Taylor et al., 2015*).

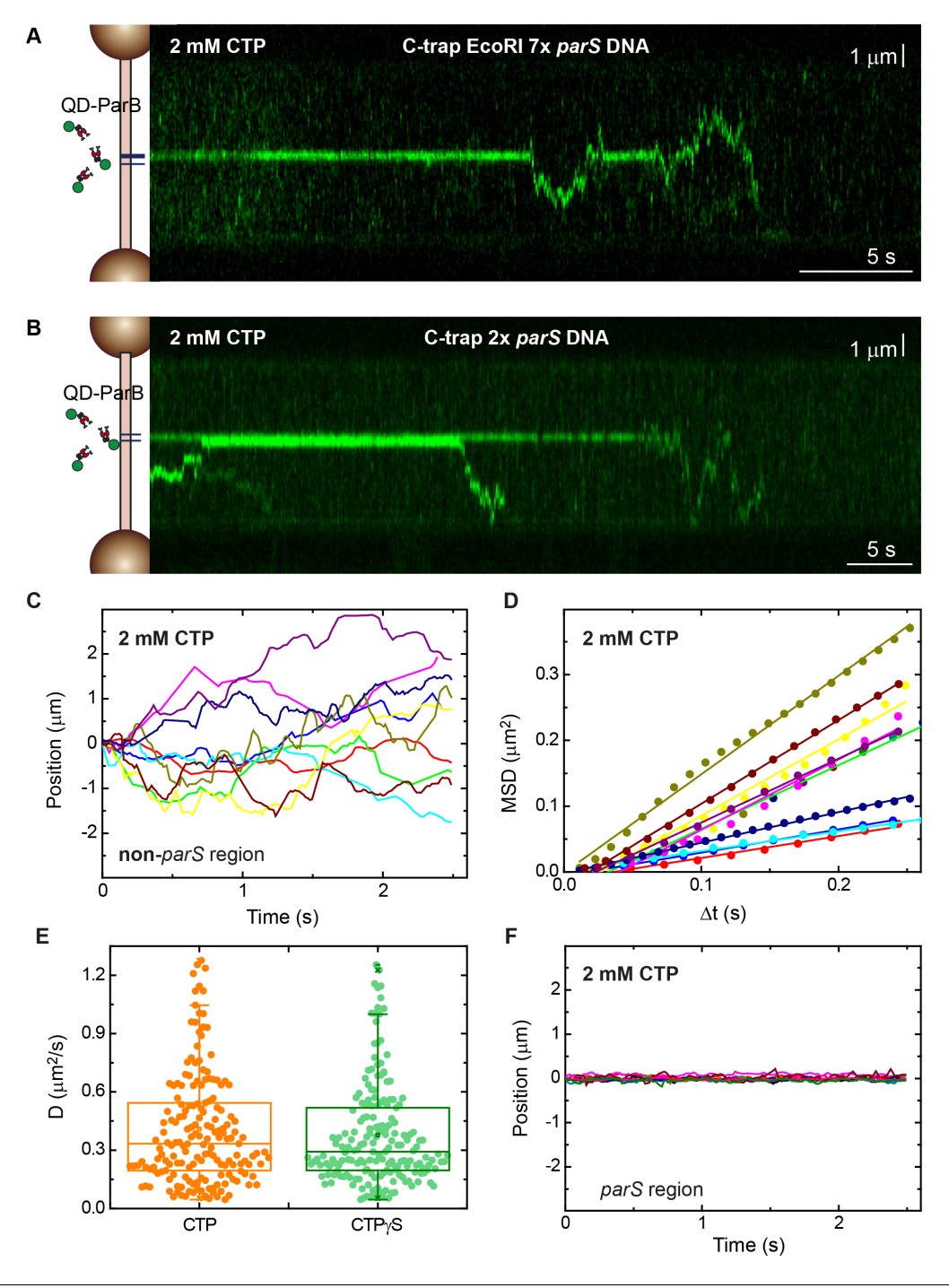

**Figure 3.** Direct visualization of ParB diffusion from parS sites. (**A**) Fluorescence kymograph of quantum dot (QD)-ParB on C-trap EcoRI 7× *parS* DNA substrates obtained under cytidine triphosphate (CTP)-Mg$^{2+}$ conditions. ParB remains mostly at *parS* sites and eventually diffuses from *parS*. (**B**) Fluorescence kymograph of QD-ParB on C-trap 2× *parS* DNA substrates obtained under CTP-Mg$^{2+}$ conditions. (**C**) Representative QD-ParB trajectories measured on non-*parS* regions of DNA (N=177). (**D**) Mean squared displacement (MSD) of ParB for different time intervals (Δt). Straight lines indicate normal diffusive behaviour. (**E**) Diffusion constants of ParB calculated as half of the slope of linear fits of MSD versus Δt. (**F**) Representative QD-ParB trajectories measured on *parS* regions of DNA indicate ParB remains mostly bound to *parS*.

The online version of this article includes the following figure supplement(s) for figure 3:

*Figure 3 continued on next page*

*Figure 3 continued*

**Figure supplement 1.** Direct visualization of ParB diffusion from *parS* sites under CTPγS conditions.

We next asked whether CTP hydrolysis is required for DNA condensation by exploiting the non-hydrolysable analogue CTPγS. We took time courses in the presence of 2 mM CTPγS and obtained force-extension curves at different ParB concentrations (*Figure 5A*). We did not observe any significant difference in force-extension measurements compared to the CTP case (*Figure 4D*). The DNA was still condensed at nanomolar ParB and condensation was abolished by EDTA (*Figure 5B*). Additional experiments using CDP showed no DNA condensation under conditions proficient for condensation (*Figure 5C*). Altogether, we conclude that binding to both CTP and $Mg^{2+}$ cofactor is required for DNA condensation by nanomolar ParB, but that CTP hydrolysis is not.

Finally, we investigated whether the dramatic stimulation of DNA condensation afforded by CTP was specific to *parS*-containing DNA and explored how condensation was affected by the number of *parS* sequences present in the substrate. Experiments using a DNA substrate with scrambled *parS* sequences did not show condensation even at $ParB_2$ concentrations of 200 nM (*Figure 6A*). Moreover, to confirm that the DNA condensation was mediated by *parS* binding, we used the mutant $ParB^{R149G}$ that cannot bind *parS* (*Autret et al., 2001*; *Fisher et al., 2017*; *Gruber and Errington, 2009*). As expected, no condensation was observed using $ParB^{R149G}$ under conditions proficient for condensation with wild-type ParB (*Figure 6B*). We next obtained force-extension curves using substrates containing different numbers of *parS* sequences. DNA molecules containing from 1 to 26 *parS* sequences and similar overall length were fabricated (*Figure 6C* and *Figure 6E*, see Materials and methods). DNA condensation was observed in substrates with 26, 13, and 7 *parS* sequences with a clear correlation between the number of *parS* and the maximum force permissive for condensation (*Figure 6D*). Experiments with substrates containing 1, 2, or 4 copies of *parS* did not result in condensation even at a relatively high $ParB_2$ concentration of 200 nM (*Figure 6E* and *Figure 6F*). We attribute this absence of condensation to a limitation of our MT assay where a minimum force needs to be applied to measure the extension of the tether. However, it has been reported that a single *parS* is sufficient for chromosome segregation and to promote ParB focus formation (*Broedersz et al., 2014*; *Jecz et al., 2015*; *Wang et al., 2017*). Therefore, we decided to employ the tethered particle motion (TPM) technique (*Kovari et al., 2018*) and atomic force microscopy (AFM; see Materials and methods) to further explore the condensation of DNA by ParB in the absence of applied force. First, we measured the root mean squared (RMS) excursions of a tethered bead in the absence of a pulling force for over 5 min. RMS values were roughly as expected for the given length of the tether (*Kovari et al., 2018*) but underwent a large decrease, consistent with DNA condensation, when we injected 200 nM $ParB_2$ and CTP-$Mg^{2+}$. Control experiments without protein or CTP or using a scrambled *parS* DNA did not reduce the RMS excursions (*Figure 6—figure supplement 1*). Moreover, AFM experiments also showed a clear interaction of ParB with single-*parS* substrates, but only when CTP was included in the reaction (*Figure 6—figure supplement 2*).

Together, these data show that CTP binding dramatically enhances DNA condensation by ParB such that it occurs efficiently at low nanomolar ParB concentration. Importantly, this stimulatory effect is completely specific for *parS*-containing DNA molecules as CTP does not improve the condensation of *parS*-free molecules that can be observed at high concentration of ParB. This presumably reflects the CTP- and *parS*-dependent recruitment of ParB sliding clamps that subsequently multimerize to effect bridging interactions between distal DNA segments.

## Discussion

We report here the first visualization of ParB binding to *parS* sequences. Previous observation of the specific binding to *parS* was hindered by the fact that ParB binding to DNA induces condensation and bridging (*Graham et al., 2014*; *Taylor et al., 2015*). To overcome this issue, we and others stretched the DNA molecules using flow or magnetic pulling combined with TIRFM (*Graham et al., 2014*; *Madariaga-Marcos et al., 2019*). However, these experiments were performed in the absence of CTP and required high concentrations of ParB, conditions which allow ParB to interact directly (i. e., from free solution rather than via *parS* loading sites) with nsDNA. The recent discovery that ParB

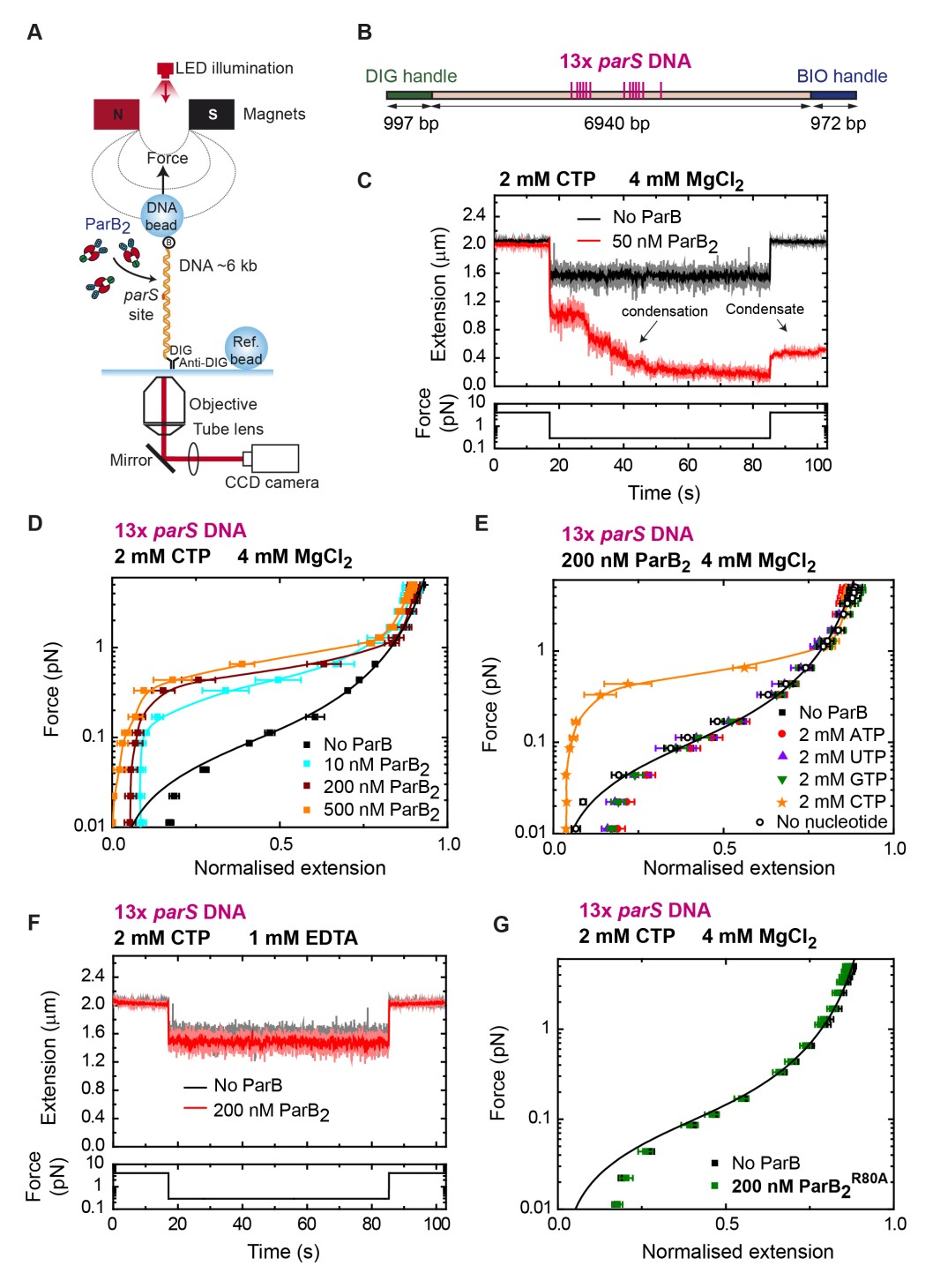

**Figure 4.** DNA condensation is induced by ParB at nanomolar concentrations in the presence of cytidine triphosphate (CTP). (**A**) Cartoon of the basic magnetic tweezers (MT) components and the layout of the experiment. (**B**) Schematic representation of the 13× *parS* DNA used for MT experiments. The positions of the *parS* sites in the DNA cartoon are represented to scale. (**C**) Condensation assay. DNA is held at 4 pN while 50 nM ParB$_2$ is injected into the fluid cell in the presence of 2 mM CTP and 4 mM MgCl$_2$. Following a 2 min incubation, the force is lowered to 0.3 pN and the extension recorded (red data). The extension in the absence of protein is shown in black. DNA could not recover the original extension by force after condensation at low force. (**D**) Average force-extension curves of 13× *parS* DNA molecules in the presence of 2 mM CTP, 4 mM MgCl$_2$, and increasing concentrations of ParB$_2$. A concentration of only 10 nM ParB$_2$ was able to condense the 13× *parS* DNA. *Figure 4 continued on next page*

*Figure 4 continued*

(**E**) Average force-extension curves of 13× *parS* DNA taken under the stated conditions and in the presence of different nucleotides or with no nucleotide. Only CTP produces condensation of *parS* DNA. Solid lines in the condensed data are guides for the eye. Errors are standard error of the mean for measurements taken on different molecules ($N \geq 6$). (**F**) Condensation assay of 13× *parS* DNA under 2 mM CTP and 1 mM EDTA conditions. DNA condensation by ParB and CTP requires $Mg^{2+}$. (**G**) CTP-binding mutant, ParB$^{R80A}$, does not condense 13× *parS* DNA under standard CTP-$Mg^{2+}$ conditions. No ParB data represent force-extension curves of DNA taken in the absence of protein and are fitted to the worm-like chain model. Errors are standard error of the mean for measurements taken on different molecules ($N = 7$).

The online version of this article includes the following source data and figure supplement(s) for figure 4:

**Source data 1.** Source data file for *Figure 4*.
**Figure supplement 1.** Low nanomolar concentrations of ParB condense *parS* DNA in the presence of cytidine triphosphate (CTP)-$Mg^{2+}$.
**Figure supplement 1—source data 1.** Source data file for *Figure 4—figure supplement 1*.

---

is a *parS*-dependent CTPase has led to the proposal of radically new models for ParB-DNA interactions in which *parS* acts as a loading site for ParB-DNA sliding clamps (*Osorio-Valeriano et al., 2019*; *Soh et al., 2019*). Together with bulk bio-layer interferometric analysis (*Jalal et al., 2020*), and supported by recent in silico modelling (*Walter et al., 2020*), this work suggests that ParB binds to *parS* in the apo state and CTP binding induces a conformational change that liberates the ParB dimer from *parS* allowing spreading. Motivated by these important new observations, we have revisited our earlier single-molecule experiments using much lower concentrations of ParB in the presence of CTP. Our results confirm many aspects of the published models but also extend them, by addressing how CTP facilitates localized DNA condensation around *parS* sites.

In an attempt to visualize the specific binding of ParB to *parS* in the presence of CTP, Soh et al. observed stretched DNA bound to a glass surface using TIRFM and reported accumulation of ParB around *parS* (*Soh et al., 2019*). Here, we used a combination of optical tweezers and confocal microscopy that allowed us not only to observe more precisely the direct binding of ParB to single DNA molecules, but also facilitated the quantification of fluorescence intensity around *parS* sites and distal non-specific sites. In our assay we found very good correlation between the position of ParB in time-averaged intensity profiles and the position of the groups of *parS* sites engineered into our substrate. These profiles easily allowed us to distinguish between the different orientations of substrate molecules (*Figure 1*). In these experiments, confocal illumination was initiated after a long ParB incubation with the substrate DNA such that the initial images should reflect equilibrium binding conditions (*Madariaga-Marcos et al., 2019*). In the absence of CTP, we observed a clear fluorescence intensity at *parS* sequences and a zero intensity (within error) in distal regions of nsDNA. We interpret this as reflecting the highly specific binding of ParB in an open clamp conformation directly to the *parS* sequences via the HtH motifs in the CDBD (*Figure 7*, step 1).

When we next added either CTP or the non-hydrolysable analogue CTPγS, we saw a four-fold increased fluorescence intensity around the *parS* sequences and a lower (but clearly non-zero) intensity in distal non-specific regions of the DNA. Note that, because this is a single-molecule experiment, the increased intensity associated with the *parS* clusters strongly suggests that there is a higher density of ParB dimers on the DNA at or near (i.e., within the spatial resolution of the imaging; ~250 nm) *parS* sequences. We interpret this as the CTP binding-dependent conversion of *parS*-bound ParB dimers into sliding clamps that move into neighbouring regions of the DNA to allow additional ParB open clamps to be recruited to the DNA (*Figure 7*, step 2). It is interesting to note that the presence of CTP and CTPγS does not prevent the initial binding of ParB to DNA by favouring a closed clamp structure *before* DNA association. This might suggest either that the DNA entry gate for ParB is a different interface (e.g., transient opening of the CTD) or simply that the CTP cannot bind efficiently until after DNA enters ParB though the NTD entry gate. In either case, following the formation of closed ParB clamps on DNA, we imagine that they spread but remain largely restrained to the region of nsDNA immediately surrounding *parS* sequences as a result of protein:protein interactions (*Figure 7*, step 2, interlaced circles). Note however that in our C-trap experiment, the ends of the DNA substrate are held at high force by the beads, meaning that the DNA cannot begin to condense by sliding into the ParB networks that are forming around *parS*. The

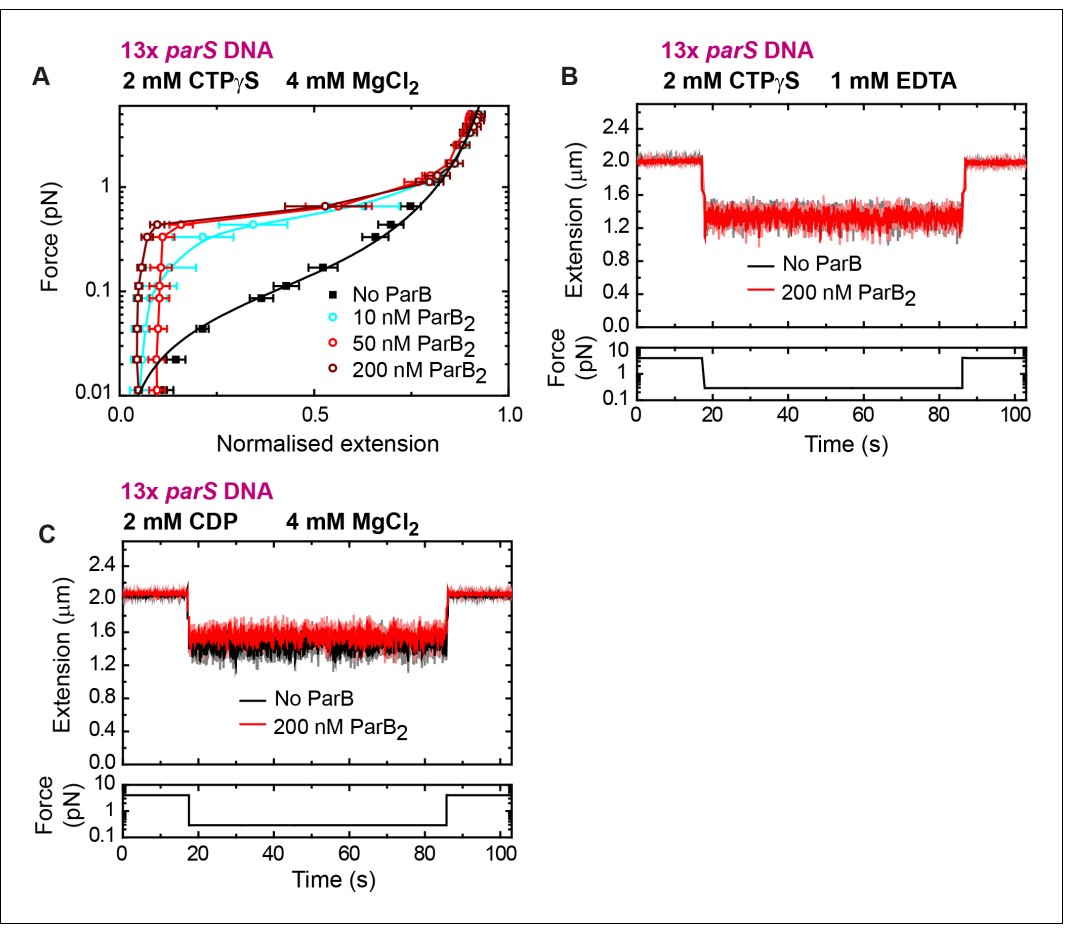

**Figure 5.** DNA condensation by nanomolar ParB requires cytidine triphosphate (CTP) binding but not hydrolysis. (**A**) Average force-extension curves of 13× *parS* DNA molecules in the presence of 2 mM CTPγS, 4 mM MgCl$_2$, and increasing concentrations of ParB. Results obtained with CTP (*Figure 4D*) and CTPγS were very similar. No ParB data represent force-extension curves of DNA taken in the absence of protein and are fitted to the worm-like chain model. Solid lines in condensed data are guides for the eye. Errors are standard error of the mean for measurements taken on different molecules ($N \geq 7$). (**B**) Condensation assay of 13× *parS* DNA under 2 mM CTPγS and 1 mM EDTA conditions. DNA condensation by ParB and CTPγS requires Mg$^{2+}$. (**C**) Condensation assay of 13× *parS* DNA under 2 mM cytidine diphosphate (CDP) and 4 mM MgCl$_2$ conditions.

The online version of this article includes the following source data for figure 5:

**Source data 1.** Source data file for *Figure 5*.

higher density of proteins (and a higher fluorescence) in the *parS* region could also arise from the design of the DNA substrate, where sliding ParBs might become trapped between adjacent *parS* sites. However, C-trap experiments using substrates containing seven or two *parS* sites (*Figure 3*) also showed higher florescence density at *parS* favouring the idea of short-range ParB network formation around *parS* versus confinement by sliding.

The weaker fluorescence intensity in distal non-specific regions of the substrate is observed exclusively in CTP or CTPγS conditions and is interpreted as ParB sliding clamps which have escaped from the ParB network at *parS* (*Figure 7*, step 2). Direct evidence that these ParB molecules are indeed involved in one-dimensional (1D) sliding interactions with the DNA is provided by experiments using QD labelled ParB (*Figure 3*). We measured a diffusion constant for ParB of 0.41±0.02 μm$^2$ s$^{-1}$, in agreement with other reports that used single-molecules analysis (*Graham et al., 2014*) and super-resolution cell imaging (*Guilhas et al., 2020*). Roadblock experiments with a catalytically inactive EcoRI mutant also supported 1D diffusion. The weaker fluorescence intensity associated with sparsely distributed ParB molecules in nsDNA regions was only observed in segments of DNA

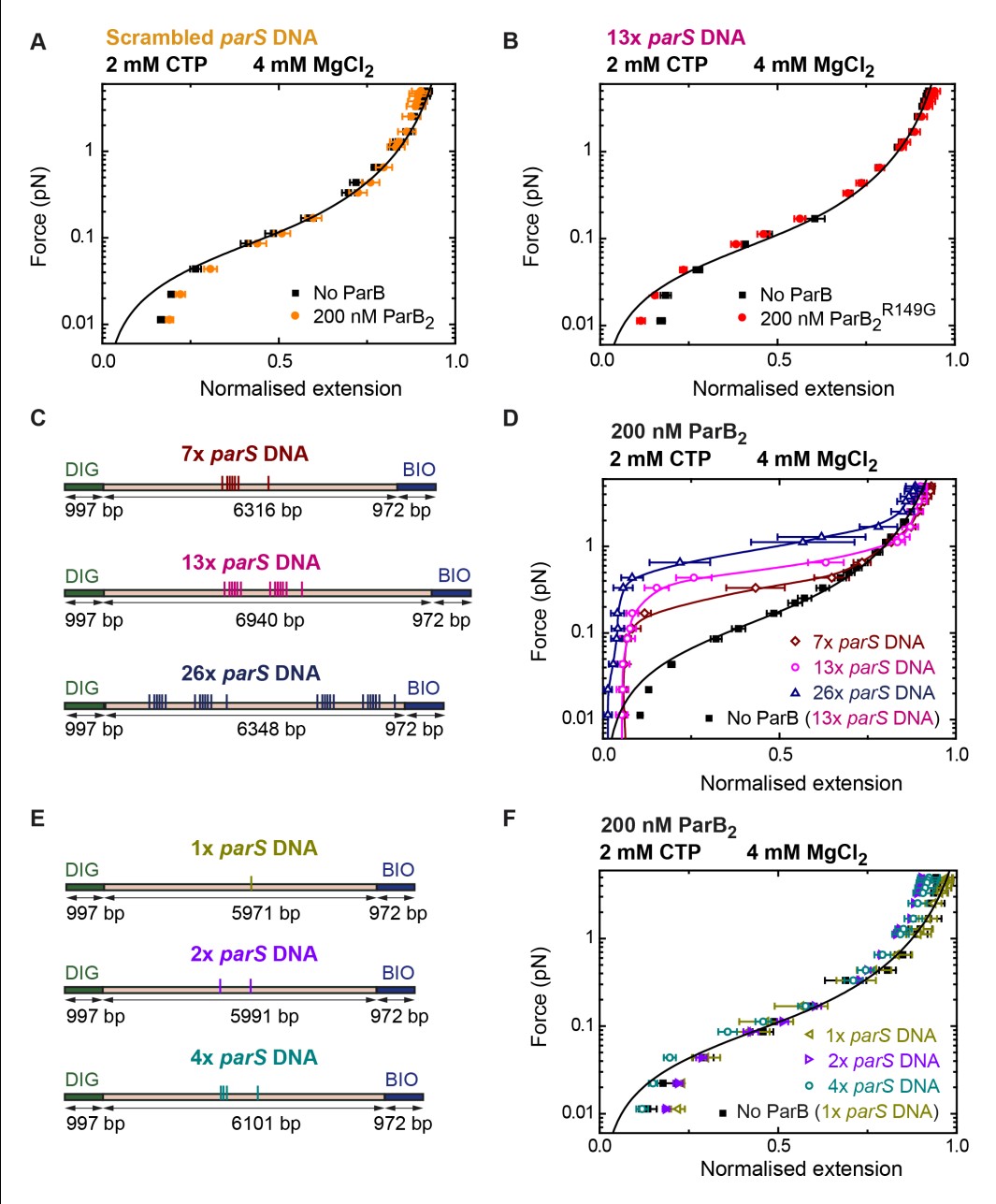

**Figure 6.** DNA condensation by nanomolar ParB is *parS* dependent. (**A**) ParB does not condense scrambled *parS* DNA under standard cytidine triphosphate (CTP)-Mg$^{2+}$ conditions. Errors are standard error of the mean of measurements on different molecules ($N$ = 5). (**B**) The *parS*-binding mutant, ParB$^{R149G}$, does not condense 13× *parS* DNA under standard CTP-Mg$^{2+}$ conditions. Errors are standard error of the mean of measurements on different molecules ($N$ = 14). (**C**) Schematic representation of DNA substrates containing 7, 13, and 26 copies of *parS*. The positions of the *parS* sites in the DNA cartoon are represented to scale. (**D**) Average force-extension curves of 7× *parS* DNA, 13× *parS* DNA, and 26× *parS* DNA obtained under standard CTP-Mg$^{2+}$ conditions. The condensation force correlates with increasing number of *parS* sequences. Solid lines in condensed data are guides for the eye. Errors are standard error of the mean of measurements on different molecules ($N \geq 7$). (**E**) Schematic representation of DNA substrates containing 1, 2, and 4 copies of *parS*. The positions of the *parS* sites in the DNA cartoon are represented to scale. (**F**) Average force-extension curves of 1× *parS* DNA, 2× *parS* DNA, and 4× *parS* DNA obtained under standard CTP-Mg$^{2+}$ conditions. No condensation was observed for these three experiments due to the pulling force present in magnetic tweezers (MT) experiments. Errors are standard error of the mean of measurements on different molecules ($N \geq 7$). No ParB data represent force-extension curves of DNA taken in the absence of protein and are fitted to the worm-like chain model.

*Figure 6 continued on next page*

*Figure 6 continued*

The online version of this article includes the following source data and figure supplement(s) for figure 6:

**Source data 1.** Source data file for *Figure 6*.
**Figure supplement 1.** Tethered particle motion (TPM) experiments show single-*parS* DNA condensation by ParB.
**Figure supplement 2.** Atomic force microscopy (AFM) experiments show single-*parS* DNA condensation by ParB.

---

containing *parS* sequences and was corralled by EcoRI roadblocks placed either side of the centromere sequences.

Despite the fact that free ParB is always present in the imaging channel in our experiments, we observed a rapid decay of the fluorescence intensity upon illumination. This suggests that the photobleaching rate is faster than the binding and unbinding kinetics of ParB. Interestingly, we observed an apparently faster fluorescence loss at distal regions of nsDNA when compared to the regions close to *parS* (*Animation 1* and *2*). This likely reflects the much higher density of ParB that our model anticipates in *parS* regions compared to distal nsDNA, where we observed the fluorescence of individual diffusing proteins. Alternatively, continuous loading at *parS* may counteract the fluorescence decay caused by photobleaching to a greater extent than at distal DNA regions (where ParB cannot bind from free solution).

In order to investigate the effects of CTP on ParB-induced DNA condensation, we employed an MT apparatus. In the presence of CTP-Mg$^{2+}$, ParB condensed *parS*-containing DNA at nanomolar concentrations, much lower than those reported before (*Fisher et al., 2017*; *Taylor et al., 2015*; *Figure 4*). Importantly, this strong stimulatory effect of CTP on DNA condensation, which is completely specific for *parS*-containing DNA, helps to resolve the question of how ParB networks can form uniquely at *parS* loci within the huge bacterial chromosome. This had not been readily apparent from previous experiments performed without CTP (see *Graham et al., 2014*; *Madariaga-Marcos et al., 2019*; *Taylor et al., 2015* for discussion). Control experiments showed that efficient condensation required both the *parS* sequence and either CTP or CTPγS, suggesting that it is CTP binding but not its hydrolysis that is important for promoting condensation. The condensation force increased with the number of *parS* sites (*Figure 6*) and a minimum value of between 5 and 7 sites was required to observe condensation, probably because a minimum force is always applied in these MT assays (see below).

We can interpret the behaviour observed in the MT using a simple extension of the model described above for the confocal microscopy experiments. In the MT experiment, the DNA ends are held apart by a very low force and the DNA is able to condense. Therefore, the ParB dimers which load around the *parS* sequence (*Figure 7*, step 2), or those which diffuse to the nsDNA region, can self-associate bringing distal DNA parts together (*Figure 7*, step 3). This creates a dynamic network of ParB molecules constraining DNA loops. Importantly, MT experiments including EcoRI roadblocks directly linked ParB diffusion with condensation because the DNA region non-accessible to ParB, due to EcoRI blocking, remained non-condensed (*Figure 7—figure supplement 1*). The greater the number of *parS* sequences present in the DNA, the greater will be the loading rate of ParB clamps into non-specific regions of the DNA, and the more stable will be the condensation against weak restraining forces. In vivo experiments have shown that a single *parS* is enough to promote chromosome segregation (*Broedersz et al., 2014*; *Jecz et al., 2015*; *Wang et al., 2017*). This might simply reflect the difference in ParB concentrations, or solution conditions that are found in vivo compared to our MT experiments. However, we favour the simplest explanation which relates to the difference in restraining forces between in vitro and in vivo conditions. Indeed, further experiments using TPM and AFM performed in the absence of force indicate that a single *parS* sequence is sufficient for DNA condensation under CTP-Mg$^{2+}$ conditions.

The molecular basis for the protein-protein interactions that hold the ParB network together in our model remains unclear and is an important subject for future study. ParB contains multiple non-exclusive interfaces within all three domains that could be relevant to this activity (*Song et al., 2017*). For example, we have previously provided direct evidence that disruption of CTD-CTD interactions decondenses DNA in vitro and prevents the formation of ParB networks in vivo, so these protein-protein interactions may contribute to network formation (*Figure 7*, step 3). However, we note that our results can also be re-interpreted in the light of the new concept of topological engagement

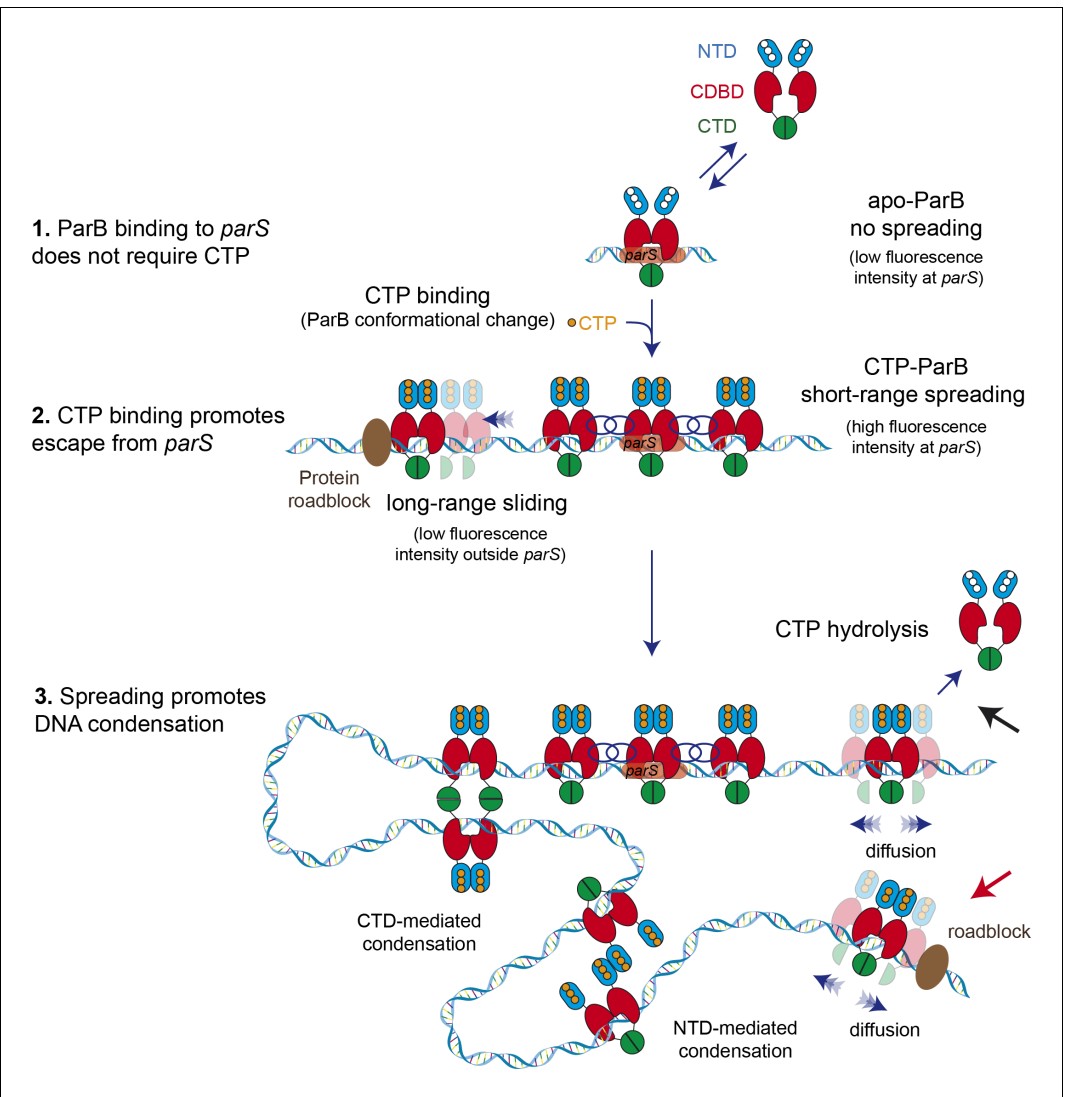

**Figure 7.** Model for ParB-dependent DNA condensation around *parS* sequences. (Step 1) ParB binding to *parS* does not require cytidine triphosphate (CTP) as observed from C-trap experiments. *ParS*-bound apo-ParB does not spread from *parS*. (Step 2) CTP binding to ParB induces a conformational change to a sliding clamp which then escapes from *parS* to neighbouring non-specific DNA. Potential interactions between the ParB proteins around *parS* are represented by interlaced blue circles. Some ParB proteins are able to slide/diffuse long distances. (Step 3) ParB spreading and diffusion promotes the interaction with other CTP-ParB dimers through the C-terminal domain (CTD) of ParB (*Fisher et al., 2017*), resulting in DNA condensation by forming large DNA loops. Alternatively, other protein-protein interaction such those mediated by the N-terminal domain (NTD) (shown in figure) or the central DNA-binding domain (CDBD) of ParB could result in DNA condensation. CTP hydrolysis might be a means to recover ParB dimers from the DNA (black arrow). Protein roadblocks constrain diffusion of ParB proteins (red arrow).

The online version of this article includes the following figure supplement(s) for figure 7:

**Figure supplement 1.** ParB diffusion is required for DNA condensation by ParB.

between DNA and a ParB toroidal clamp. If CTD-CTD interactions are important for closing the sliding clamp around DNA as has been suggested (*Jalal et al., 2020*; *Soh et al., 2019*), then disruption of these interactions would dissolve ParB networks, not by breaking protein-protein interactions between ParB molecules, but rather by releasing the constrained DNA loops and promoting ParB dissociation into free solution. Other possible interfaces that may establish ParB networks include those that have been observed between the CDBD or NTD (*Chen et al., 2015*; *Leonard et al.,*

*2004*; *Schumacher and Funnell, 2005*). A final open question concerns the loading of the SMC complexes at bacterial centromeres which leads to the individualization of bacterial chromosomes following DNA replication (*Hayes and Barillà, 2006*; *Schumacher, 2008*). Despite the intimate functional relationship between the ParABS partitioning system and SMC complexes, the nature of the physical interactions between these systems and their regulation are not understood.

## Materials and methods

### Protein preparation

WT-ParB, R149G ParB, and AF488-ParB were prepared as described (*Fisher et al., 2017*; *Madariaga-Marcos et al., 2019*). An expression construct for R80A ParB was generated by site-directed mutagenesis of the wild-type expression plasmid (QuikChangeII XL, Agilent Technologies, Santa Clara, CA). The mutant protein was expressed and purified in the same manner as wild type. The EcoRI E111G variant was a gift from Michelle Hawkins (University of York) and was prepared as described previously (*King et al., 1989*). All quoted concentrations of ParB refer to ParB dimers ($ParB_2$).

To prepare biotinylated-ParB (Bio-ParB), the frozen protein stock of ParB S68C (the same that was used to be labelled with Alexa Fluor 488 $C_5$ Maleimide; *Fisher et al., 2017*; *Madariaga-Marcos et al., 2019*) was defrosted and buffer exchanged into storage buffer minus glycerol by loading the protein into a Superose 6 Increase 10/300 GL gel filtration column. The protein was then treated with 0.02 mM TCEP for at least 30 min at 4°C before labelling with 10-fold excess of EZ-Link Maleimide-$PEG_{11}$-Biotin disolved in DMSO (Thermo Fisher Scientific, Waltham, MA). The labelling reaction was performed overnight at 4°C with end-over-end rotation. The following day, reaction was quenched by adding DTT at a 5 mM final concentration for 2 hr at 4°C. The excess of Maleimide-$PEG_{11}$-Biotin was removed by size exclusion using the Superose 6 Increase 10/300 GL column. Before glycerol addition, the labelling efficiency was estimated using the HABA (4'-hydroxyazobenzene-2-carboxylic acid) method with the Pierce Biotin Quantitation Kit (Thermo Fisher Scientific, Waltham, MA), indicating that the most protein was efficiently labelled. The CTPase activity of Bio-ParB was comparable to wild-type ParB (data not shown).

### Bio-ParB QD conjugation

Bio-ParB was incubated with streptavidin-coated QD (Qdot 525 Streptavidin Conjugate, Thermo Fisher Scientific, Waltham, MA) in a molar ratio of 1:4 for 5 min in ice and diluted to a final concentration of 10 nM Bio-ParB QD in reaction buffer (100 mM NaCl, 50 mM Tris-HCl pH 7.5, 4 mM $MgCl_2$, 1 mM DTT, and 0.1% Tween-20). The sample was then incubated for 10 additional minutes on ice with 10 µM biotin to block the remaining streptavidin-free binding sites and supplemented with 2 mM CTP, as required.

### Fabrication of DNA plasmids with multiple copies of *parS*

DNA plasmids containing multiple *parS* sequences (optimal sequence of *B. subtilis parS* = 5′-TGTTCCACGTGAAACA) were produced by modification of the plasmids described in *Taylor et al., 2015*, where the cloning of a plasmid containing a 'scrambled' *parS* site (scrambled *parS*: 5'-**C**GTG**CC**CA**G**GG**A**G**A**CA; bold represents mutated nucleotides) was also reported.

Plasmids with increasing number of *parS* sequences were produced as follows. First, we annealed two long oligonucleotides (*Supplementary file 2*) containing two *parS* sites separated by a single XbaI restriction site. The oligonucleotides were hybridized by heating at 95°C for 5 min and cooled down to 20°C at a −1 °C min$^{-1}$ rate in hybridization buffer (10 mM Tris-HCl pH 8.0, 1 mM EDTA, 200 mM NaCl, and 5 mM $MgCl_2$). These oligonucleotides were designed to create an incomplete XbaI site at both ends after ligation, so that once ligated to a cloning plasmid they cannot be cut again by XbaI. The single bona fide XbaI site located in the middle of the oligonucleotide insert allows repetition of the ligation process in the cloning plasmid as many times as desired to add new pairs of *parS* sequences. Plasmids containing 1× *parS*, 2× *parS*, 4× *parS*, and 7× *parS* were obtained following this procedure. Inserts for subsequent cloning of plasmids containing 13× *parS*, 26× *parS*, and 39× *parS* were produced by PCR using a high-fidelity polymerase (Phusion Polymerase, NEB) (see *Supplementary file 2* for primer sequences). Plasmids were cloned in DH5α competent cells

(Thermo Fisher Scientific, Waltham, MA) and purified from cultures using a QIAprep Spin Miniprep Kit (QIAGEN). All plasmids were checked by DNA sequence analysis. A detailed description of these procedures is included below and in *Figure 1—figure supplement 2*.

All constructs were cloned into the vector pET28a(+) (Novagen). Plasmids were cloned in DH5α Competent cells (Thermo Fisher Scientific, Waltham, MA), and after selection of possible positive colonies by colony PCR, plasmids were purified from cultures using QIAprep Spin Miniprep Kit (QIA-GEN), analyzed by restriction digestion, and finally checked by DNA sequence analysis. These plasmids were employed to prepare different MT substrates which sequences are included in *Supplementary file 1*.

### 'Scrambled' *parS* DNA
The cloning of a plasmid containing a 'scrambled' *parS* site (scrambled *parS*: 5'-**C**GT**G**CC**CA**GG**G**A-**G**ACA) was described in *Taylor et al., 2015*.

### 1× *parS* DNA
The cloning of a plasmid containing a 1× *parS* sequence (optimal sequence of *B. subtilis parS* = 5'-TGTTCCACGTGAAACA) was also described in *Taylor et al., 2015*.

### 2× *parS* DNA
The plasmid containing 2× *parS* sequences was derived from the 1× *parS* plasmid by ligation of annealed synthetic oligonucleotides containing the *parS* sequence with appropriate overhangs into the cut NcoI site.

### 4× *parS* DNA
Two long 5'-phosphorylated oligonucleotides (*Supplementary file 2*) containing 2× *parS* sites separated by an XbaI restriction site and ending in a modified XbaI restriction site were hybridized as described above. The plasmid containing 2× *parS* sequences was digested with XbaI, dephosphory-lated with rSAP (NEB), and ligated with this pair of hybridized oligonucleotides rendering a plasmid with 4× *parS* sites.

The oligonucleotides were designed to create an incomplete XbaI site at both ends after ligation in such a way that once ligated, XbaI cannot cut again in those positions. In addition, in the middle of the annealed oligonucleotides, a bona fide XbaI site was included, allowing the repetition of the ligation process to insert the annealed oligonucleotides as many times as desired to increase the number of copies of the *parS* site (*Figure 1—figure supplement 2*).

### 7× *parS* DNA
The plasmid containing 4× *parS* sequences was digested with XbaI, dephosphorylated and ligated again with the pair of hybridized oligonucleotides rendering a plasmid with 6× *parS* sites (not employed in this paper). However, in this step of cloning and by chance, the pair of hybridized oligo-nucleotides entered two times during the ligation process. Although that ligation should render a plasmid with 8× *parS* sites, notice that one of the *parS* sites was incomplete (TTTCACGTGGAACA) probably because an error in the synthesis of one of the oligonucleotides, and therefore that incom-plete sequence was not considered as a *parS* site and the final construct is said to contain 7× *parS* sequences.

### 13× *parS* DNA
The plasmid containing 7× *parS* sequences was used as a template to amplify by PCR (Phusion Poly-merase, NEB) the six *parS* fragment by using primers 32.F pET28 PCR NdeI and 33.R pET28 PCR NdeI (*Supplementary file 2*). The PCR fragment was digested with NdeI, and ligated into the plas-mid with 7× *parS* sequences previously digested with NdeI and dephosphorylated, rendering a plas-mid with 13× *parS* sites.

### 26× *parS* DNA and 39× *parS* DNA

The plasmid containing 13× *parS* sequences was used as a template to amplify by PCR the 13 *parS* fragment by using 50.F pET28 37 SphI-BglI and 51.R pET28 BglI-SphI (*Supplementary file 2*). The PCR fragment was digested with BglI, and ligated into the plasmid with 13× *parS* sequences previously digested with BglI and dephosphorylated, rendering a plasmid with 26× *parS* sites. In addition, in this step of cloning and by chance, in a different plasmid the PCR fragment entered two times during the ligation process, and therefore we also obtained a plasmid with 39× *parS* sites. This last one was employed to prepare large plasmids as described below.

## Fabrication of large plasmids (> 17 kbp) for C-trap experiments

Long fragments of DNA (>17 kbp) containing a custom sequence for C-trap experiments were fabricated as follows. A large plasmid was formed by ligation of three pieces. Two of them correspond to two PCR-fabricated DNAs (C1 and C2) derived from lambda DNA (NEB) as template and including suitable restriction sites in the designed primers (*Supplementary file 2*). The third part (C3) containing the sequence of interest (e.g., multiple copies of the *parS* sequence) was produced by plasmid digestion. The three fragments were then ligated and DH5α competent cells transformed by regular heat shock procedure. Large plasmids were purified from cultures using QIAprep Spin Miniprep Kit and checked by DNA sequence analysis. A detailed description of these procedures is included below and in *Figure 1—figure supplement 3*.

Plasmids containing the *parS* region flanked by two clusters of 5× EcoRI restriction sites were produced following the same procedure but replacing parts C1 and C2 with fragments C1-EcoRI and C2-EcoRI, each one including a cluster of 5× EcoRI restriction sites at the desired position (*Supplementary file 2*).

### C-trap 39× *parS* DNA

The large plasmid containing 39× *parS* sites was obtained by ligation of three fragments of DNA (*Figure 1—figure supplement 3*). Two of them correspond to two PCR-fabricated DNAs (C1 and C2) derived from lambda DNA (NEB) as template and including suitable restriction sites in the designed primers (*Supplementary file 2*). Fragment C1 (green), that corresponds to positions 33464–38474 bp of lambda DNA, was amplified with oligos 20Lambda_F_NotI and 21Lambda_R_-SalI, purified and digested with NotI and SalI. Fragment C2 (black), that corresponds to positions 5475–14509 bp of lambda DNA, was amplified with oligos 22Lambda_F_BamHI and 23Lambda_R_-NotI, purified and digested with NotI and BamHI. The fragment containing the 39× *parS* sites (C3, brown in the scheme) was obtained by digestion of the plasmid containing 39× *parS* sites described above with BamHI and SalI, dephosphorylation and purification by gel extraction with Gel Extraction Kit (QIAGEN). The three fragments were then ligated at a ratio 1:1:1 and DH5α Competent cells were transformed by regular heat shock procedure. After selection of possible good colonies, large plasmids were purified from cultures using QIAprep Spin Miniprep Kit and checked by restriction digestion followed by DNA sequence analysis.

### C-trap EcoRI 39× *parS* DNA

The large plasmid containing the 39× *parS* region flanked by two regions of 5× EcoRI restriction sites (named as C-trap EcoRI 39× *parS*) was produced following the same procedure but replacing parts C1 and C2 with fragments C1-EcoRI and C2-EcoRI, each one including five EcoRI restriction sites at the desired position. However, during the PCR amplification of fragment C2-EcoRI with oligos 177.Lambda_F_5Eco BamHI and 23Lambda_R_NotI, the forward oligo was annealed in a different position of the lambda DNA due to a similar sequence, and therefore the amplified fragment was shorter and corresponds to positions 13254–14509 bp of lambda DNA.

### C-trap EcoRI 7× *parS* DNA

The large plasmid containing the 7× *parS* region flanked by two regions of 5× EcoRI restriction sites (named as C-trap EcoRI 7× *parS* DNA) was produced by ligating C1-EcoRI and C2-EcoRI to a C3 fragment obtained by digestion of 7× *parS* DNA with BamHI and SalI.

## C-trap 2× parS DNA

The large plasmid containing 2× parS sites (named as C-trap 2× parS DNA) was produced as C-trap 39× parS DNA, but using as C3 fragment the digestion product of 2× parS DNA with BamHI and SalI.

## Fabrication of small plasmids (≤3.6 kbp) for AFM experiments

Plasmids employed to prepare different AFM substrates, which sequences are included below, were cloned into the vector pBlueScriptIISK+ (Stratagene), because of its smaller size. All substrates were checked by DNA sequence analysis.

## AFM no parS DNA

This plasmid corresponds to the commercial pBlueScriptIISK+ vector that does not contain any parS site.

## AFM 1× parS DNA

The plasmid containing 1× parS sequence was used as a template to amplify by PCR the AFM 1× parS PCR fragment by using 74.F-TPM OK NotI and 46.R post SalI pET oligonucleotides (*Supplementary file 2*). The PCR fragment was digested with NotI (rendering a fragment with NotI and blunt ends), and ligated into the plasmid pBlueScriptIISK+, previously digested with PspOMI and PsiI.

### MT DNA substrates

MT DNA substrates were produced as described in *Taylor et al., 2015* and essentially consist of a central part (~6 kbp) containing different number of parS sequences or a non-specific scrambled parS site, flanked by two smaller fragments (~1 kbp) labelled with biotins or digoxigenins used as immobilization handles. Handles for MT substrates were prepared by PCR (see *Supplementary file 2* for primers) including 200 µM final concentration of each dNTP (G,C,T,A) and 10 µM Bio-16-dUTP or Dig-11-dUTP (all from Roche). Labelled handles specifically bind either to a glass surface covered with anti-digoxigenins or to superparamagnetic beads covered with streptavidin. About 40% of molecules fabricated using this procedure were torsionally constrained in MT experiments. Sequences of the central part of MT substrate are included in Supplementary methods section.

### C-trap DNA substrates

C-trap DNA substrates consisted of a large central part of 17–25 kbp containing 39, 7, or 2 copies of the parS sequence, flanked or not by two clusters of 5× EcoRI restriction sites produced by linearization of large plasmids with NotI (NEB). Without further purification, the fragment was ligated to highly biotinylated handles of ~1 kbp ending in *NotI*. Handles for C-trap substrates were prepared by PCR (see *Supplementary file 2* for primers) including 200 µM final concentration of each dNTP (G,C,A), 140 µM dTTP, and 66 µM Bio-16-dUTP. These handles were highly biotinylated to facilitate the capture of DNA molecules in C-trap experiments. As both sides of the DNA fragment end in *NotI*, it is possible to generate tandem (double-length) tethers flanked by the labelled handles. Sequences of the central part of C-trap substrates are included in Supplementary methods section.

A control C-trap DNA substrate based on lambda DNA was prepared according to a previously described protocol (*Wasserman et al., 2020*) with slight modifications; 10 nM lambda DNA was incubated with 33 µM each of dGTP, dATP, biotin-16-dUTP and biotin-14-dCTP (Thermo Fisher Scientific, Waltham, MA), and 5 units of DNA polymerase (Klenow Fragment [3'→5' exo-], NEB) in 1× NEB2 buffer for 1 hr at 37˚C. Reaction was followed by heat inactivation of the enzyme for 20 min at 75˚C. Sample was ready to use in C-trap experiments without further purification. DNAs were never exposed to intercalanting dyes or UV radiation during their production and were stored at 4˚C.

### TPM substrates

TPM DNA substrates consisted of a short central fragment (~1.7 kbp) containing a single parS sequence or a non-specific 'scrambled' parS site, flanked by oligonucleotides labelled with either multiple digoxigenins or biotins. Each central fragment was PCR-amplified using the 1× parS or the

'scrambled' *parS* DNA plasmid as template, respectively, and including suitable restriction enzyme sites in the designed primers (*Supplementary file 2*). Labelled oligonucleotides were fabricated based on a previously published method (*Camunas-Soler et al., 2013*; *Marin-Gonzalez et al., 2020*). Briefly, 27P-XhoI-A and 24P-NotI-A oligonucleotides (*Supplementary file 2*) were biotin or digoxigenin tailed using terminal transferase (NEB) and either BIO-dUTP or DIG-dUTP, respectively. The modified oligonucleotides were purified using a QIAquick nucleotide removal kit (Qiagen) and hybridized respectively with 26XhoI-B or 25NotI-B (*Supplementary file 2*). The central fragment was digested with XhoI and NotI enzymes and ligated overnight to the two hybridized tailed oligonucleotides using T4 DNA ligase (NEB). Any excess of oligonucleotides was removed using a Microspin S-400 column (Cytiva).

## AFM DNA substrates

AFM DNA substrates consist of a small plasmid (3557 bp) containing a single *parS* sequence, or a control, pBlueScriptIISK+ plasmid (2961 bp, Stratagene) without *parS*. Details of fabrication and sequences are included in Supplementary methods section.

## NTP hydrolysis measurement by Malachite green colorimetric detection

A pair of oligonucleotides containing a *Scrambled parS* site, 1× *parS* or 2× *parS* sites (see *Supplementary file 2* for sequences) were hybridized by heating at 95℃ for 5 min, and cooled down to 20℃ at a $-1$ ℃ min$^{-1}$ rate in hybridization buffer (10 mM Tris-HCl pH 8.0, 1 mM EDTA, 200 mM NaCl, and 5 mM MgCl$_2$). Mixtures of NTP (2×) and DNA (2×) in reaction buffer (100 mM NaCl, 50 mM Tris-HCl pH 7.5, 4 mM MgCl$_2$, 1 mM DTT, and 0.1% Tween-20) were prepared on ice. Protein solutions (2×) containing either wild-type ParB or ParB$^{AF488}$ in reaction buffer were also prepared on ice. NTP/DNA pre-mix (5 μl) was added to protein solution (5 μl) and mixed on ice. Phosphate standards and blanks were prepared in parallel for each experiment. After mixing, samples containing 1 mM NTP, 0.5 μM DNA, 8 μM ParB$_2$ were placed in a PCR machine set to 25℃ for 30 min. Additionally, different concentrations (0.25, 0.75, and 1 μM) of a DNA with two *parS* sequences were tested in presence of 1 mM CTP. Samples were diluted by the addition of 70 μl water, then mixed with 20 μl working reagent (WR) (Sigma, Ref MAK 307) and transferred to a flat-bottom 96-well plate. The plate was incubated for 30 min at 25℃ and the absorbance was measured at a wavelength of 620 nm in a SpectraMax iD3 (VERTEX Technics) plate reader that uses the SoftMax Pro7 software. Readings were performed in rounds to preserve the same 30 min WR incubation time for all samples. Absorbance values from the phosphate standard samples were corrected with the absorbance for 0 μM phosphate. Absorbance values from the ParB samples were corrected by the reaction buffer absorbance (blank). Absorbance values from the phosphate standard samples were used to plot an OD$_{620\ nm}$ versus pmol phosphate standard curve. All samples were tested in duplicate. ParB$^{AF488}$ retains *parS*-stimulated CTPase activity within twofold levels of wild-type protein (*Figure 1—figure supplement 1B and C*).

## MT experiments

MT assays were performed using a home-made setup similar to an apparatus that has been described previously (*Carrasco et al., 2013*; *Kemmerich et al., 2016*; *Pastrana et al., 2016*; *Strick et al., 1998*). Briefly, optical images of micrometer-sized superparamagnetic beads tethered to a glass surface by DNA constructs are acquired with a 100× oil-immersion objective and a CCD camera. Real-time image videomicroscopy analysis determines the spatial coordinates of the beads with nm accuracy in the x, y, and z directions (*Pastrana et al., 2016*). A step motor located above the sample moves a pair of magnets allowing the application of stretching forces to the bead-DNA system. Applied forces can be quantified from the Brownian excursions of the bead and the extension of the DNA tether (*Strick et al., 1998*). Data were acquired at 120 Hz to minimize sampling artefacts in force determination. We used vertically aligned magnets coupled to an iron holder to achieve a force of up to 5 pN.

DNA was diluted and mixed in ParB-Mg$^{2+}$ buffer (50 mM Tris-HCl [pH 7.5], 100 mM NaCl, 0.2 mg ml$^{-1}$ BSA, 0.1% Tween-20, 1 mM DTT, 4 mM MgCl$_2$) or ParB-EDTA buffer (by replacing the 4 mM MgCl$_2$ with 1 mM EDTA) and then incubated with 1 μm diameter magnetic beads (Dynabeads, Myone Streptavidin, Invitrogen, Carlsbad, CA) for 10 min. Magnetic beads were previously washed

three times and diluted in PBS. DNA:beads ratios were adjusted for each substrate to obtain single tethered beads. Following DNA-beads incubation, then sample was injected in a double PARAFILM (Sigma)-layer flow cell. After 10 min adsorption of the beads to the surface, we applied a force of 4 pN to remove the non-attached beads and washed with buffer to clean the chamber. Torsionally constrained molecules and beads with more than a single DNA molecule were identified from its distinct rotation-extension curves (*Gutierrez-Escribano et al., 2020*) and discarded for further analysis.

Time course data was obtained by recording the extension of the tether at a low force of 0.33 pN, after a 2 min incubation of DNA tethers and ParB at 4 pN. MT experiments using the EcoRI roadblocks include an initial step of incubation with 200 nM EcorRI$^{E111G}$ for 10 min at 4 pN. To obtain force-extension curves, we measured the extension of the tethers at decreasing forces from 5 to 0.01 pN for a total measuring time of around 15 min. Force-extension curves were first measured on naked DNA (no ParB data) and always from high to low force. Then, the experiment was repeated on the same molecule but at quoted ParB$_2$ concentrations. This method allowed us to obtain force-extension curves in the absence and presence of protein for each tethered DNA molecule. No ParB DNA curves were fitted to the worm-like chain model using Origin Software. Molecules with a discrepancy of contour length of ±15% from the crystallographic length were discarded for the analysis.

## C-trap fluorescence experiments

We used a dual optical tweezers setup combined with confocal microscopy and microfluidics (C-trap) from Lumicks (Lumicks B.V.). Our system has three laser lines (488, 532, 535 nm) for confocal microscopy and provides a force resolution of <0.1 pN at 100 Hz, distance resolution of <0.3 nm at 100 Hz, and confocal scanning with < 1 nm spot positioning accuracy (Lumicks). In this work, we used a 488 nm laser for illumination and a 500–525 nm filter for its fluorescence. We used a five-channel microfluidic chamber (*Figure 1—figure supplement 1A*). Channel 1 contained 4.38 μm SPHERO streptavidin-coated polystyrene beads diluted at 0.005% w/v in fishing buffer (FB, 10 mM Tris-HCl pH 8.0, and 50 mM NaCl). Channel 2 included the 39× *parS* DNA in FB and channel 3 only FB. First, two beads were trapped using the dual optical tweezers in channel 1 and moved to channel 2 to attach DNA molecules to the beads. The capture of DNA was detected by performing a preliminary force-extension curve in channel 2. Then, the bead-DNA system was moved to channel 3, where further force-extension curves were performed to check for single-molecule captures and to define the zero force point. Finally, a stretching force of 19–23 pN was set, and the bead-DNA system moved to the protein channel 4 that contains ParB$_2^{AF488}$ at quoted concentrations in ParB-Mg$^{2+}$ buffer supplemented with an oxygen scavenger system (1 mM Trolox, 20 mM glucose, 8 μg ml$^{-1}$ glucose oxidase, and 20 μg ml$^{-1}$ catalase). Confocal images (scans) and kymographs were performed in the protein channel 4 (*Figure 1—figure supplement 1A*), which was previously passivated with BSA (0.1% w/v in PBS) and Pluronic F128 (0.5% w/v in PBS).

Spreading experiments include a 2 min incubation time in channel 4 before turning on the confocal laser. Spreading-blocking experiments also used four channels but in this case we inject 100 nM EcoRI$^{E111G}$ in ParB buffer in channel 3. Following a 2 min incubation in channel 3 to allow binding of the blocking protein, the bead-DNA system was moved to channel 4, which in this case contained 100 nM EcoRI$^{E111G}$ and 20 nM ParB$_2^{AF488}$ in ParB buffer supplemented with the oxygen scavenger (see above). An additional 2 min incubation was performed before confocal imaging.

Confocal images of a defined area were taken using a pixel size of 100 nm and a scan velocity of 1 μm ms$^{-1}$. With these parameters, typical images of experiments using single or tandem 39× *parS* DNA were obtained every 2 and 2.7 s, respectively. Confocal laser intensity at the sample was 3.4 μW.

Kymographs were obtained by single-line scans between the two beads using a pixel size of 100 nm and a pixel time of 0.1 ms. Typical temporal resolution of kymographs taken on single or tandem 39× *parS* DNA were 25 and 32 ms, respectively.

Both biological (new sample preparations from a fresh stock aliquot) and technical (MT and C-trap experiments) repeats were undertaken.

## C-trap data analysis

Data acquisition was carried out using Bluelake, the commercial software included in the Lumics C-trap. This software provides HDF5 files, which can be processed using Lumics' Pylake Python package. We used home-made Python scripts to export the confocal scans or kymographs in ASCII matrix files or in TIFF format. Python scripts can be found at https://github.com/Moreno-Herrero-Lab/C-TrapDataProcessing (copy archived at swh:1:rev:12e7d7f36053cb872fd53c0c5c5b9cab8e304835; *Moreno-Herrero and de Bragança, 2021*). Profiles were obtained from ASCII files using a home-written LabView program. Images of scans or kymographs were produced using the WSxM freeware (*Horcas et al., 2007*). Animated GIFs were produced using Image J from individual scans saved in TIFF.

## Measurement of diffusion constants from C-trap kymographs

Kymographs of individual trajectories of QD-ParB were obtained at ~20 ms time resolution and analysed using a home-written LabView program to extract the protein position along the DNA as a function of time. The length of the time courses was restricted to 2.5 s to increase the statistical sample. The diffusion constant (*D*) was calculated for each individual trajectory using the same analysis program from the slope of a linear fit of 1D MSD taken at different time intervals (*t*) (*Equation 1*; *Gorman and Greene, 2008*; *Heller et al., 2014*). For each experimental condition, more than 177 trajectories were considered:

$$\mathrm{MSD} = 2\mathrm{Dt} + \mathrm{offset} \tag{1}$$

## TPM experiments

We mixed TPM 1× *parS* DNA or TPM scrambled *parS* DNA substrates (1717 bp) with 1 µm beads following the same procedure as described for MT experiments. Then, we injected the sample into the fluid chamber of our MT setup, from which we had previously removed the magnet head to prevent the application of pulling forces to the beads. Following the attachment of tethers and extensive rinse to remove unbound particles, we then tracked in-plane coordinates of single beads using the MT tracking software. We computed the excursions of a bead using *Equation 2*:

$$\mathrm{RMS} = \sqrt{(x-\bar{x})^2 + (y-\bar{y})^2} \tag{2}$$

where $x$ and $y$ are the in-plane coordinates of the bead and $\bar{x}$ and $\bar{y}$ the average for the measured time. To reduce the potential effect of drift in the signal, we also computed the square root of the sum of the variances of particle position over a time window as described (*Equation 3*; *Han et al., 2009*):

$$\mathrm{RMS}_\tau = \sqrt{\left\langle (x-\bar{x})^2 + (y-\bar{y})^2 \right\rangle_\tau} \tag{3}$$

where $\tau$ is the duration of the time window of the filter and in this case, $\bar{x}, \bar{y}$ are averages for the given time window. The optimal time window for filtering $\tau$ was selected to be longer than the characteristic time of spatial correlation of the DNA-bead system considering the calibration routine described in *Han et al., 2009*. The value depends on the contour length and the radius of the bead and for our experimental system we found a minimum time window for filtering of $\tau \sim$ 8 s. Thus, we filtered the data to 8.3 s. For each individual DNA tether, we obtained a mean $\mathrm{RMS}_\tau$ over the whole time of the experiment of 300 s.

## AFM experiments

ParB$_2$ was diluted to a concentration of 10 nM in the protein buffer (50 mM Tris HCl pH 7.5, 100 mM NaCl, 4 mM MgCl$_2$) and incubated with 0.3 nM DNA and 3.33 mM CTP for 5 min at RT. The same protocol was followed but without adding CTP in control experiments. Then, MgCl$_2$ was added to a final concentration of 7.5 mM up to a final volume of ~20 µl and the sample deposited onto freshly cleaved mica. The sample was rinsed with 2 ml of Milli-Q water and dried with nitrogen gas. Samples of plasmids without protein contained 0.3 nM DNA in the same protein buffer and 7.5 mM MgCl$_2$. Images were taken in tapping mode in air, using an AFM from Nanotec Electronica S.L. with

PointProbePlus tips (PPPNCH Nanosensors). Images were processed using the WSxM software (*Horcas et al., 2007*).

## Acknowledgements

We are grateful to Michelle Hawkins (University of York) for supplying the EcoRI E111G variant. Work in the MSD lab was supported by the Wellcome Trust (New Investigator Award 100401/Z/12/Z to MSD) and the BBSRC (South West Biosciences Doctoral Training studentship to GLMF). FM-H acknowledges support from the European Research Council (ERC) under the European Union Horizon 2020 Research and Innovation Program (grant agreement 681299). Work in the Moreno-Herrero laboratory was also supported by Spanish Ministry of Economy and Competitiveness grant BFU2017-83794-P (AEI/FEDER, UE to FM-H) and Comunidad de Madrid grants Tec4-Bio – S2018/NMT-4443 and NanoBioCancer – Y2018/BIO-4747 (to FM-H).

## Additional information

### Funding

| Funder | Grant reference number | Author |
| --- | --- | --- |
| European Research Council | 681299 | Fernando Moreno-Herrero |
| Wellcome Trust | 100401/Z/12/Z | Mark Simon Dillingham |
| BBSRC | South West Biosciences Doctoral Training studentship | Gemma LM Fisher |
| MINECO | BFU2017-83794-P | Fernando Moreno-Herrero |
| Comunidad de Madrid | Tec4Bio – S2018/NMT-4443 | Fernando Moreno-Herrero |
| Comunidad de Madrid | NanoBioCancer - Y2018/BIO-4747 | Fernando Moreno-Herrero |

The funders had no role in study design, data collection and interpretation, or the decision to submit the work for publication.

### Author contributions

Francisco de Asis Balaguer, Carried out all C-trap, MT, and TPM experiments, and performed data analysis. Critically reviewed the manuscript and approved the final version; Clara Aicart-Ramos, Writing - review and editing, Produced all DNA substrates and carried out NTPase assays. Produced ParB proteins. Critically reviewed the manuscript and approved the final version; Gemma LM Fisher, Produced ParB proteins. Critically reviewed the manuscript and approved the final version; Sara de Bragança, Produced Python scripts for analysis of C-trap data. Critically reviewed the manuscript and approved the final version; Eva M Martin-Cuevas, Carried out AFM experiments. Critically reviewed the manuscript and approved the final version; Cesar L Pastrana, Developed methods for magnetic tweezers and TPM measurements. Critically reviewed the manuscript and approved the final version; Mark Simon Dillingham, Wrote the paper. Conceived the project and designed experiments. Critically reviewed the manuscript and approved the final version; Fernando Moreno-Herrero, Wrote the first draft of the manuscript, designed experiments, developed software for data analysis, conceived and supervised the project. Critically reviewed the manuscript and approved the final version

### Author ORCIDs

Francisco de Asis Balaguer  https://orcid.org/0000-0002-6393-2863
Clara Aicart-Ramos  https://orcid.org/0000-0002-1114-4259
Sara de Bragança  http://orcid.org/0000-0003-4039-1993
Cesar L Pastrana  https://orcid.org/0000-0002-6988-8146
Mark Simon Dillingham  https://orcid.org/0000-0002-4612-7141
Fernando Moreno-Herrero  https://orcid.org/0000-0003-4083-1709

**Decision letter and Author response**
Decision letter https://doi.org/10.7554/eLife.67554.sa1
Author response https://doi.org/10.7554/eLife.67554.sa2

## Additional files

### Supplementary files
- Supplementary file 1. Sequences of DNA fragments used in this work.
- Supplementary file 2. DNA oligonucleotides used in this work.
- Transparent reporting form

### Data availability
All DNA sequences used are included in Supplementary Information. Data sets for Fig. 1E, 1F; Fig. 1-S1B, Fig.1-S1C; Fig.1-S4C, Fig1-S4D; Fig. 2C, 2G; Fig. 4C, 4D, 4E, 4F, 4G; Fig. 4-S1A, Fig. 4-S1B, Fig. 4-S1C; Fig. 5A, 5B; Fig. 6A, 6B, 6D, 6F, have been provided.

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

## Appendix 1

### Supplemental information

This paper contains supplemental information.

