## [Decision Letter]

**Acceptance summary:**

This study applies tour de force combination of single-molecule approaches including the correlative optical tweezers – fluorescence microscopy, magnetic tweezers, tethered particle motion and AFM to examine the mechanism of assembly of bacterial partition/segregation complexes by the ParB protein and the ensuing condensation of *parS*-containing DNA. The experiments visualized how this assembly and its DNA specificity are promoted by CTP. The authors convincingly show that following association at *parS*, CTP-binding allowed ParB to diffusively spread along the DNA. ParB spreading along the DNA was in turn the prerequisite for DNA condensation mediated by this protein. This study will be of broad interest to those studying protein-DNA interactions and cell division. The nanomechanical DNA condensation experiments together with the combined direct fluorescent visualization represent a helpful methodological development for future studies of this and similar systems.

**Decision letter after peer review:**

Thank you for submitting your article "CTP promotes efficient ParB-dependent DNA condensation by facilitating one-dimensional spreading from parS" for consideration by *eLife*. Your article has been reviewed by 4 peer reviewers, and the evaluation has been overseen by Maria Spies as a Reviewing Editor and Cynthia Wolberger as the Senior Editor. The reviewers have opted to remain anonymous.

Essential Revisions:

While the reviewers and the reviewing editor agreed that this is a very thorough, carefully done and important study, they identified a number of issues that need to be addressed:

1. All reviewers agree that, if it is possible with a reasonable effort, the authors should provide some direct single-molecule diffusion data, as the presented data support the proposed behavior, but do not rule out all alternatives

2. The data show that multiple parS sites are needed for DNA condensation in the presented experiments, while in *B. subtilis* a single parS site appears sufficient for all functions, potentially implying that DNA condensation is not important. The authors openly acknowledge this and offer two explanations (page 10), including differences in ParB concentration and solution conditions. The conclusions of the manuscript will be significantly strengthen if the authors can vary these parameters and identify conditions that may allow (limited) DNA condensation with a single parS site.

3. Address the additional comments in individual reviews.

*Reviewer #1:*

This study investigates how the recently discovered CTPase activity of bacterial ParB promotes the separation of ParS-containing sister chromosomes within the ParABS system. Using fluorescence visualization the authors were directly visualizing ParB-binding to ParS as well as the diffusive spreading of the protein along the DNA. They furthermore systematically and comprehensively probed under which conditions ParB-mediated DNA condensation was obtained that is considered to be essential for the chromosome segregation. The data of the authors convincingly shows that CTP binding by ParB stimulates its association with ParS and most importantly promotes its diffusive spreading from ParS. Diffusive spreading along the DNA contour was demonstrated using protein roadblocks. This provides sufficient evidence for protein diffusion along the DNA contour, although the authors did not achieve to visualize the diffusion of single ParB complexes along DNA directly. By testing a broad range of conditions (ParB concentration, presence of Mg^2+^, hydrolyzable and non-CTP, number of ParS sites), the authors demonstrated that CTP binding rather than hydolysis is sufficient for ParB to promote DNA condensation. The two different types of observation show together that the CTP-mediated diffusive spreading of ParB from ParS drives the downstream DNA condensation. Moreover, the nanomechanical DNA condensation experiments together with the combined direct fluorescent visualization represent a helpful methodological development for future studies of this system. Overall, the presented work is a clear and comprehensive study that provides direct and unambiguous evidence for the recently suggested models of ParB-mediated DNA condensation and its stimulation by ParS and CTP.

1) Given the potential of the employed C-trap setup, it would be highly interesting to directly see the 1D diffusion of single ParB complexes along the DNA. Could the authors obtain such data with ease and if not, what was hindering the single-molecule observation? Knowledge about the diffusion constant/friction coefficient could potentially help to understand how freely diffusing complexes may promote DNA condensation.

2) Assuming the ParB complexes rapidly diffuse along DNA, how do the authors imagine that condensation would be promoted by such proteins that lack any anchor (see also 1)? Could the interaction with other ParB complexes upon looping quench the mobility of the ParB complexes? Please add a short discussion.

3) The authors show the CTP binding rather than hydrolysis is required to promote the diffusive spreading and the DNA condensation. The role of CTP hydrolysis remains therefore unclear. I could imagine that hydrolysis may stimulate the release of ParB from DNA and this way potentially the decondensation forces of condensed DNA. Did the authors test such a scenario?

4) The authors should also provide a fluorescence profile for the absence of CTP in Figure 1C supplement 4.

*Reviewer #2:*

This manuscript investigates parS DNA binding and condensation by *B. subtilis* ParB protein in single molecule experiments using optical traps and magnetic tweezers. The work follows up on the recent discovery of ParB's ability to bind and hydrolyze CTP. The authors show that CTP addition stimulates ParB binding to DNA molecules harboring clustered parS sites and promotes the spreading of ParB onto neighboring DNA. Moreover, ParB binding is shown to lead to the condensation of DNA with clustered parS sites, again an activity that is stimulated by CTP or the non-hydrolysable analog CTPgS.

While the work carefully observes and describes ParB activity in vitro and reports findings that are largely consistent with recent publications, a potential weakness of the study concerns the use of artificially clustered parS sites on the DNA test substrates and the absence of similar activity on more natural substrates with a single parS site, together raising doubts about the physiological relevance of the discoveries.

parS sites are widely dispersed on the *B. subtilis* genome (10 sites in ~1 Mb) with the two closest sites being separated by about 10 kb. Moreover, a single parS site is sufficient to support chromosome segregation and to promote normal ParB focus formation. Clearly, parS site clustering is not crucial in vivo. Yet, the activities detected in this work appear strictly dependent on parS clustering. It is thus not clear whether DNA condensation by ParB as described here is indeed occurring in vivo and if it is relevant for ParB function.

A related preprint (Taylor et al., 2021) has reported similar DNA condensation using F plasmid ParB. In this case however clustered parS sites are naturally found on the corresponding DNA, the F plasmid. DNA condensation has thus only been observed with clustered parS sites. This should be discussed and further investigated.

The authors claim that they can directly detect binding of ParB to parS ('first visualization of the specific binding of ParB to parS'). However, their assay does not discriminate between parS-bound ParB and 1D diffusing ParB. This issue is particularly important as recent studies have actually suggested that ParB only transiently binds parS during the CTP hydrolysis cycle (Jalal et al., 2020; Soh et al., 2019). Can the authors discriminate parS binding and sliding by using a single parS site and EcoRI roadblocks?

ParB-AF488 displays severely compromised CTPase activity (figure supplement 1). Is this due to the cysteine mutation or due to the fluorescent label? The results need to be confirmed using a differently labeled protein (different cysteine or different fluorophore depending on what causes the reduced activity).

*Reviewer #3:*

ParBs bind to a centromere site called parS to form large condensed complexes at which ParBs are observed to spread many bp away from parS, but the mechanisms of spreading are under debate. The current study is based on recent discoveries that ParBs bind and hydrolyse CTP, and that CTP promotes spreading. Here the authors directly visualize fluorescent ParBs bound to parS on DNA that is stretched out because it is tethered at both ends. They examine the condensation using magnetic tweezers on DNA tethered at one end and pulled by a magnet at the other end. They find that CTP does promote parS-specific DNA binding and spreading, and this activity requires CTP but not hydrolysis. The results extend their previous published analyses in the absence of CTP, in which the authors observed spreading but it was not parS-specific and required higher ParB concentrations. Their results recapitulate many of the properties of spreading that have been observed in vivo, including specificity for parS and the influence of roadblocks. The results are consistent with a model proposed by Soh et al. (2019) in which CTP locks ParB as a clamp around DNA by promoting N-terminal domain self association, and that once clamped, it slides along the DNA away from parS; that is, sliding is proposed to be the mechanism of spreading. The results however are also consistent with spreading by cooperative interactions of ParBs with those bound next to them, so these data do not directly support sliding by ruling out other alternatives. Since this distinction is not resolved by the current results, one could look at these results as confirmatory. However there is important value to these results. As the authors state, this is the first time ParB molecule binding on linear DNA at and around parS has been directly visualized in single molecule studies; that is, with resolution for parS vs closely surrounding regions. The ability to view the complexes directly at this resolution on the DNA directly tests ParB/parS localization and the influence of spreading roadblocks. Second, the specificity for parS (or lack thereof) has long been a problem for the in vitro study of ParB binding to parS in biophysical experiments. Also, the authors show that sliding still prefers to reside close to the parS region (it is not "free"), suggesting that lateral (and perhaps bridging) protein-protein interactions play roles in complex architecture.

1. Figure 2G: The appropriate control is the same experiment omitting EcoRI, which should be included. Otherwise we are forced to compare different DNA substrates with this one and from previous figures.

2. Page 7 top: "suggests that it occurs by sliding from parS". Although the authors clearly favor the sliding clamp model (as do I), their data do not address whether sliding or lateral protein-protein recruitment is responsible for spreading in these experiments. The key difference is whether the ParB molecules that have spread to neighboring DNA were loaded at parS or just next to other ParBs in what they call "short-range spreading". Roadblocks also are consistent with sliding but other models can accommodate them also.

3. Page 2 and elsewhere: "an absolute requirement for CTP binding but not hydrolysis". The authors did not test CDP, although Soh et al. showed that CDP does promote similar N-terminal self association that is promoted by CTP. This is an important part of the model; whether CDP works or whether the protein must be in an apo form to dissociate from DNA. Have the authors tested CDP? If yes, these data could be included. If not, CDP should be discussed.

4. Figure 1-S1C: the right 3 bars look mislabeled because they are identical to the middle 3 bars as labeled. Are they with CTPgS?

5. Figure 1-S4: what is time=0? Turning on the excitation?

*Reviewer #4:*

Francisco et al. investigate the role of CTP and hydrolysis in the binding of ParB to parS sequence and non-specific DNA at the single-molecule level. Using optical tweezers, they show the specific binding of ParB to parS sites, and demonstrate that this process is enhanced by the presence of CTP or CTPγS. They find that lower density ParB proteins are also detected in distal non-specific DNA in the presence of parS, and that ParB spreading is restricted by protein roadblocks. Furthermore, using magnetic tweezers, they show that parS-containing DNA molecules are condensed by ParB at nanomolar protein concentration, which requires CTP binding but not hydrolysis. These finding show the significance of CTP-dependent ParB spreading and impact the understanding of the mechanism of DNA bridging and condensation by ParB networks.

Based on these results, the authors propose a model for ParB-mediated DNA condensation, which requires one-dimensional ParB sliding along DNA from parS sites. Overall, the experiments were carefully done and thoroughly controlled. The manuscript provides critical insights that can be strengthened by addressing the following concerns:

1. Did the authors observe the diffusion of isolated ParB foci along DNA? This will provide strong evidence for the proposed diffusion/sliding model.

2. Based on the sliding clamp model, ParB spreading and diffusion result in DNA condensation by forming large DNA loops. Is it possible to show the dynamic spreading of ParB while keep the same numbers of ParB on DNA? For example, can the authors incubate ParB-containing DNA in channel 4 (ParB channel) at a certain time for the loading of ParB on parS sites, and then move it to the buffer channel without free ParB as well as with CTP or CTPγS, where the images are acquired at the long interval time to minimize the photobleaching. The fluorescent intensity of the ParB during the spreading process can be analyzed. If the intensity remains constant through spreading in the presence of CTPγS but significantly decrease in the presence of CTP, this data will strongly demonstrate the proposed spreading and CTP hydrolysis-dependent dissociation mechanism.

3. In Figure 2, the authors show the spreading of ParB can be blocked by EcoRI. Can the authors show that EcoRI is bound at the specificity positions? The spreading blockage by protein roadblocks showed in optical tweezers experiments potentially hints that the roadblocks may affect the DNA condensation. Can the authors apply the magnetic tweezers to show the affection of protein roadblocks to DNA condensation in vitro?

---

## [Author Response]

Essential Revisions:While the reviewers and the reviewing editor agreed that this is a very thorough, carefully done and important study, they identified a number of issues that need to be addressed:1. All reviewers agree that, if it is possible with a reasonable effort, the authors should provide some direct single-molecule diffusion data, as the presented data support the proposed behavior, but do not rule out all alternatives.

To answer this query, we fabricated a biotinylated version of ParB to be labelled with QD-streptavidin. QD labelling increased the fluorescence intensity of a single ParB protein and allowed much longer visualization times due to the absence of photobleaching. We also fabricated substrates containing only 7 or 2 copies of parS to increase our chances of observing individual events. We succeeded in observing one-dimensional diffusion of individual ParB molecules along the DNA. Kymographs also showed a clear stable binding at parS sites confirming our previous findings. We developed new software to analyse these kymographs and extract individual ParB time-courses (i.e. ParB position along DNA versus time). Furthermore, we calculated diffusion constants for individual trajectories. The ParB diffusion constant was 0.41±0.02 μm2 s^-1^ (mean±sem, n=177) or 3.5±0.2 kbp2 s^-1^, considering a rise per base pair of 0.34 nm. These data are consistent with previous single molecule experiments performed in the absence of CTP (Graham et al., 2014) and with the recently calculated ParB diffusion constant of 0.7 μm2 s^-1^ using super-resolution microscopy (Guilhas et al., 2020).

These new data directly confirm our conclusion of ParB diffusion from parS sites. Moreover, we now provide a measurement of ParB diffusion constant in CTP and CTPγS conditions. Additionally, we also show that, often, ParB proteins leave the DNA from the non-specific region.

All these new results have been included in a new Results section entitled “Direct visualization of ParB diffusion from parS sequences” and a new Figure 3 and Figure 3 —figure supplement 1, which caused relabelling of subsequent figures. New sections in the Methods regarding biotinylation of ParB, conjugation with QDs, and calculation of diffusion constants were also included in the revised manuscript.

2. The data show that multiple parS sites are needed for DNA condensation in the presented experiments, while in *B. subtilis* a single parS site appears sufficient for all functions, potentially implying that DNA condensation is not important. The authors openly acknowledge this and offer two explanations (page 10), including differences in ParB concentration and solution conditions. The conclusions of the manuscript will be significantly strengthen if the authors can vary these parameters and identify conditions that may allow (limited) DNA condensation with a single parS site.

We have now addressed the question over the number of parS sequences required for condensation in several ways. First, we have fabricated C-trap substrates containing only 7 or 2 copies of parS and repeated the C-trap experiments with these new substrates. Essentially, we reproduced the same results as with 39x parS DNAs and our diffusion experiments were done using these new substrates (see also above).

Our data show a direct correlation between condensation forces and the number of parS sequences in MT experiments. As noted by the reviewer, we observe that a minimum number of parS are required to observe condensation in MT experiments. We attribute this to the fact that these experiments are always done in the presence of a pulling force, which might be sufficient to prevent condensation if the condensation force is low. In order to test this hypothesis, we have now also performed experiments in the absence of pulling forces using the tethered particle motion (TPM) method and atomic force microscopy (AFM).

We first used TPM to determine RMS excursions of the bead, a parameter correlated with the contour length of the DNA. We made a DNA molecule suitable for TPM experiments of ~1700 bp containing a single parS. In the presence of CTP, we observed a strong decrease of RMS excursions consistent with DNA condensation. The same experiment done without CTP, or with a scrambled parS DNA did not reduce RMS excursions. These experiments are included in a new Figure 6 supplement 1.

Additionally, we used AFM to image circular DNA molecules containing zero or one parS sites incubated under different experimental conditions. In the absence of CTP, DNA molecules and ParB proteins appeared monodispersed on the mica surface. In the presence of CTP and only when the single-parS DNA substrate was used, we observed clustering and condensation of DNA regions, presumably the parS region. These experiments are included in a new Figure 6 supplement 2. Together these experiments confirm that ParB interactions lead to DNA condensation and that a single parS is sufficient for DNA condensation.

New text has been added to the manuscript explaining these experiments. Additional methods and figures have also been added.

3. Address the additional comments in individual reviews.

We have addressed the additional comments of individual reviewers below.*Reviewer #1:*

This study investigates how the recently discovered CTPase activity of bacterial ParB promotes the separation of ParS-containing sister chromosomes within the ParABS system. Using fluorescence visualization the authors were directly visualizing ParB-binding to ParS as well as the diffusive spreading of the protein along the DNA. They furthermore systematically and comprehensively probed under which conditions ParB-mediated DNA condensation was obtained that is considered to be essential for the chromosome segregation. The data of the authors convincingly shows that CTP binding by ParB stimulates its association with ParS and most importantly promotes its diffusive spreading from ParS. Diffusive spreading along the DNA contour was demonstrated using protein roadblocks. This provides sufficient evidence for protein diffusion along the DNA contour, although the authors did not achieve to visualize the diffusion of single ParB complexes along DNA directly. By testing a broad range of conditions (ParB concentration, presence of Mg^2+^, hydrolyzable and non-CTP, number of ParS sites), the authors demonstrated that CTP binding rather than hydolysis is sufficient for ParB to promote DNA condensation. The two different types of observation show together that the CTP-mediated diffusive spreading of ParB from ParS drives the downstream DNA condensation. Moreover, the nanomechanical DNA condensation experiments together with the combined direct fluorescent visualization represent a helpful methodological development for future studies of this system. Overall, the presented work is a clear and comprehensive study that provides direct and unambiguous evidence for the recently suggested models of ParB-mediated DNA condensation and its stimulation by ParS and CTP.1) Given the potential of the employed C-trap setup, it would be highly interesting to directly see the 1D diffusion of single ParB complexes along the DNA. Could the authors obtain such data with ease and if not, what was hindering the single-molecule observation? Knowledge about the diffusion constant/friction coefficient could potentially help to understand how freely diffusing complexes may promote DNA condensation.

We completely agree with the reviewer, and in fact, we had been working on this aim while the paper was under review. Please see our detailed answer to this query in the “essential revisions” section elaborated by the editor.

In the view of these new diffusion data, it becomes clear that in MT experiments, ParB proteins are present around the non-specific region of the DNA substrate. However, the high restraining force prevents them from intermolecular interactions. We envision that the reduction of the force brings distal DNA segments which are stabilized through ParB-ParB interactions, leading to DNA condensation.

2) Assuming the ParB complexes rapidly diffuse along DNA, how do the authors imagine that condensation would be promoted by such proteins that lack any anchor (see also 1)? Could the interaction with other ParB complexes upon looping quench the mobility of the ParB complexes? Please add a short discussion.

Our data support a model where ParB proteins diffuse from parS sites. This should result in the presence of multiple ParBs on the non-specific DNA section, while the molecule is held at forces non-permissive for condensation (as we see in C-trap experiments, Figure 1 and Figure 2). Further decrease of the force (as we do in MT experiments) will bring distant proteins together, resulting in the formation of loops ultimately leading to the condensation of DNA. Of course, this model of condensation requires intermolecular interactions, which might come from the CTD or the NTD of ParB, but we do not see the necessity for proteins to be anchored at non-specific sites. Ultimately, modelling could bring light to this point, but this is out of the scope of this work.

We have added some text to the discussion regarding this point.

3) The authors show the CTP binding rather than hydrolysis is required to promote the diffusive spreading and the DNA condensation. The role of CTP hydrolysis remains therefore unclear. I could imagine that hydrolysis may stimulate the release of ParB from DNA and this way potentially the decondensation forces of condensed DNA. Did the authors test such a scenario?

Our new C-trap experiments show that often ParB detaches from nonspecific DNA after diffusing from parS sites. This is consistent with previous models proposing that it is the hydrolysis of CTP which promotes the release of ParB from DNA. We tried to measure ParB lifetimes on non-specific DNA on CTP and CTPγS conditions but our results were non-conclusive, and this is the subject for future work. Measurement of de-condensation forces in MT experiments is complicated because we found that those depend on the level of previous compaction and the potential nonspecific attachment of the condensate to the glass surface.

4) The authors should also provide a fluorescence profile for the absence of CTP in Figure 1C supplement 4.

We have now added a time-averaged profile for the absence of CTP in Figure 1C supplement 4.

Reviewer #2:This manuscript investigates parS DNA binding and condensation by *B. subtilis* ParB protein in single molecule experiments using optical traps and magnetic tweezers. The work follows up on the recent discovery of ParB's ability to bind and hydrolyze CTP. The authors show that CTP addition stimulates ParB binding to DNA molecules harboring clustered parS sites and promotes the spreading of ParB onto neighboring DNA. Moreover, ParB binding is shown to lead to the condensation of DNA with clustered parS sites, again an activity that is stimulated by CTP or the non-hydrolysable analog CTPgS.While the work carefully observes and describes ParB activity in vitro and reports findings that are largely consistent with recent publications, a potential weakness of the study concerns the use of artificially clustered parS sites on the DNA test substrates and the absence of similar activity on more natural substrates with a single parS site, together raising doubts about the physiological relevance of the discoveries.parS sites are widely dispersed on the *B. subtilis* genome (10 sites in ~1 Mb) with the two closest sites being separated by about 10 kb. Moreover, a single parS site is sufficient to support chromosome segregation and to promote normal ParB focus formation. Clearly, parS site clustering is not crucial in vivo. Yet, the activities detected in this work appear strictly dependent on parS clustering. It is thus not clear whether DNA condensation by ParB as described here is indeed occurring in vivo and if it is relevant for ParB function.

To address the point of parS clustering, we fabricated two new C-trap substrates containing 7 and 2 copies of parS (new Figure 2 supplement 1A, new Figure 3, and new Figure 3 supplement 1). Essentially, we obtained the same results as with the 39x parS, discarding an effect of parS clustering. Experiments with QD-labelled ParB showed stable binding to single parS sites and allowed us to directly observe diffusion of ParB proteins along non-specific DNA. Further experiments using the TPM technique (new Figure 6 supplement 1), and AFM (new Figure 6 supplement 2) with substrates with a single parS confirmed CTP-dependent DNA condensation. Therefore, all these new experiments indicate that parS clustering is not required for condensation.

A related preprint (Taylor et al., 2021) has reported similar DNA condensation using F plasmid ParB. In this case however clustered parS sites are naturally found on the corresponding DNA, the F plasmid. DNA condensation has thus only been observed with clustered parS sites. This should be discussed and further investigated.

We have observed a direct correlation between condensation forces and the number of parS sequences in MT experiments. We also observed that a minimum number of parS were required to observe condensation in the MT experiments. We attribute this to the fact that these experiments are always done in the presence of a pulling force, which might be sufficient to prevent condensation if the condensation force is low. In order to test this hypothesis, we performed experiments in the absence of pulling forces using the tethered particle motion (TPM) method and atomic force microscopy (AFM) techniques. performed experiments in the absence of pulling forces using the tethered particle motion (TPM) method and atomic force microscopy (AFM) techniques.

We first used the TPM method to determine RMS excursions of the bead, a parameter correlated with the contour length of the DNA. We fabricated a DNA molecule suitable for TPM experiments of ~1700 bp containing a single parS. In the presence of CTP we observed a strong decrease of RMS excursions consistent with DNA condensation. The same experiment done without CTP, or with a scrambled parS DNA did not reduce RMS excursions. These experiments are included in a new Figure 6 supplement 1.

Additionally, we used AFM to image circular DNA molecules with or without a single parS site incubated under different experimental conditions. In the absence of CTP, DNA molecules and ParB proteins appeared monodispersed on the mica surface. In the presence of CTP and only when the single-parS DNA substrate was used, we observed clustering and condensation around a region of DNA, presumably the parS region. These experiments are included in a new Figure 6 supplement 2.

Together these experiments confirm that ParB interactions lead to DNA condensation and that a single parS is enough for DNA condensation.

New text has been added to the manuscript explaining these experiments. Additional methods and figures have also been added.

The authors claim that they can directly detect binding of ParB to parS ('first visualization of the specific binding of ParB to parS'). However, their assay does not discriminate between parS-bound ParB and 1D diffusing ParB. This issue is particularly important as recent studies have actually suggested that ParB only transiently binds parS during the CTP hydrolysis cycle (Jalal et al., 2020; Soh et al., 2019). Can the authors discriminate parS binding and sliding by using a single parS site and EcoRI roadblocks?

This is an important point, also raised by the other reviewers. Please see our detailed answer to this query in the “essential revisions” section elaborated by the editor.

ParB-AF488 displays severely compromised CTPase activity (figure supplement 1). Is this due to the cysteine mutation or due to the fluorescent label? The results need to be confirmed using a differently labeled protein (different cysteine or different fluorophore depending on what causes the reduced activity).

We agree with the reviewer that the ParB-AF488 has a reduced CTPase activity as we have transparently reported in our original submission. This reduction is not explained by errors in protein quantification due to the presence of the label, as spectroscopic and gel-based methods for estimating protein concentration are in good agreement (data not shown). It may be that a fraction of the protein is inactivated by the long labelling procedure (the reviewer is comparing labelled protein with untreated wild-type protein, rather than mock-labelled protein).

In any case, the apparent effect on CTPase and DNA binding is less than 2-fold (original paper and Author response image 1), and DNA binding stimulates the CTPase activity to the same extent as for wild type (original paper). These data suggest that labelling causes minimal disruption to the enzyme’s active sites. Therefore, we respectfully disagree that the activity of this protein is “severely compromised” by labelling. We do accept and acknowledge that the behavior is perturbed from wild type as is the case in many single molecule studies using fluorescently-labelled proteins.

Importantly, this small quantitative change in CTP turnover rate has no effect on our conclusions. Our work focuses on the *large and qualitative* effects of CTP on the protein’s behaviour, and these are consistent across experiments which use both labelled (C-trap) and unlabeled (MT, TPM, AFM) protein preparations. In all cases, we control for these CTP-dependent effects by performing the experiments in the absence of nucleotides or in the presence of non-hydrolysable analogues or CDP.

**Author response image 1. sa2fig1:** Representative TBM- and TBE-EMSAs assessing *parS* binding (A and B) and non-specific DNA-binding activity (C), respectively, of Alexa-488-labelled-ParB and its precursor, ParBS68C. Wild-type-like specific and non-specific DNA-binding activity is retained. In-gel detection of the fluorescent protein corresponds to the pattern of nucleoprotein complexes.

Reviewer #3:ParBs bind to a centromere site called parS to form large condensed complexes at which ParBs are observed to spread many bp away from parS, but the mechanisms of spreading are under debate. The current study is based on recent discoveries that ParBs bind and hydrolyse CTP, and that CTP promotes spreading. Here the authors directly visualize fluorescent ParBs bound to parS on DNA that is stretched out because it is tethered at both ends. They examine the condensation using magnetic tweezers on DNA tethered at one end and pulled by a magnet at the other end. They find that CTP does promote parS-specific DNA binding and spreading, and this activity requires CTP but not hydrolysis. The results extend their previous published analyses in the absence of CTP, in which the authors observed spreading but it was not parS-specific and required higher ParB concentrations. Their results recapitulate many of the properties of spreading that have been observed in vivo, including specificity for parS and the influence of roadblocks. The results are consistent with a model proposed by Soh et al. (2019) in which CTP locks ParB as a clamp around DNA by promoting N-terminal domain self association, and that once clamped, it slides along the DNA away from parS; that is, sliding is proposed to be the mechanism of spreading. The results however are also consistent with spreading by cooperative interactions of ParBs with those bound next to them, so these data do not directly support sliding by ruling out other alternatives. Since this distinction is not resolved by the current results, one could look at these results as confirmatory. However there is important value to these results. As the authors state, this is the first time ParB molecule binding on linear DNA at and around parS has been directly visualized in single molecule studies; that is, with resolution for parS vs closely surrounding regions. The ability to view the complexes directly at this resolution on the DNA directly tests ParB/parS localization and the influence of spreading roadblocks. Second, the specificity for parS (or lack thereof) has long been a problem for the in vitro study of ParB binding to parS in biophysical experiments. Also, the authors show that sliding still prefers to reside close to the parS region (it is not "free"), suggesting that lateral (and perhaps bridging) protein-protein interactions play roles in complex architecture.1. Figure 2G: The appropriate control is the same experiment omitting EcoRI, which should be included. Otherwise we are forced to compare different DNA substrates with this one and from previous figures.

Following the reviewer’s suggestion, we have performed a control experiment without the EcoRI mutant (Author response image 2). As expected, no blocking was observed. Note that including the profile in Figure 2G is not appropriate because the molecules are different, with different fluorescence intensities and orientations. We have included a sentence in the main text indicating this result as not shown.

**Author response image 2. sa2fig2:** ParB spreading in the absence of EcoRI roadblocks using C-trap EcoRI 39x *parS* DNA.

2. Page 7 top: "suggests that it occurs by sliding from parS". Although the authors clearly favor the sliding clamp model (as do I), their data do not address whether sliding or lateral protein-protein recruitment is responsible for spreading in these experiments. The key difference is whether the ParB molecules that have spread to neighboring DNA were loaded at parS or just next to other ParBs in what they call "short-range spreading". Roadblocks also are consistent with sliding but other models can accommodate them also.

This is an important point, also raised by the other reviewers. Please see our detailed answer to this query in the “essential revisions” section elaborated by the editor.

3. Page 2 and elsewhere: "an absolute requirement for CTP binding but not hydrolysis". The authors did not test CDP, although Soh et al. showed that CDP does promote similar N-terminal self association that is promoted by CTP. This is an important part of the model; whether CDP works or whether the protein must be in an apo form to dissociate from DNA. Have the authors tested CDP? If yes, these data could be included. If not, CDP should be discussed.

Following the reviewer’s suggestion, we performed C-trap and MT experiments using CDP. Our results showed ParB binding to parS at levels similar to the no-nucleotide condition (Author response image 3), and lack of diffusion from parS. As expected, no condensation of DNA was observed under CDP conditions (Author response image 3). This confirms our conclusion that it is the binding of CTP and not the hydrolysis what enhances ParB binding to parS, and promotes ParB diffusion from parS and condensation.

**Author response image 3. sa2fig3:** (left) C-trap scan showing ParB binding to *parS* at similar levels to the Apo-form of ParB, and not diffusion from *parS*. (right) MT time-course showing absence of condensation in the presence of CDP and Mg^2+^.

We have included the fluorescence intensity profile corresponding to the CDP condition in Figure 2C. We have also included the MT experiment showing the absence of condensation as a new Figure 5C. These new results are described and commented in the main text.

4. Figure 1-S1C: the right 3 bars look mislabeled because they are identical to the middle 3 bars as labeled. Are they with CTPgS?

Thank you for pointing out this typo. Indeed it should, and now does, read CTPγS.

5. Figure 1-S4: what is time=0? Turning on the excitation?

Yes, time = 0 corresponds to turning on the excitation. We have included this sentence in the caption of Figure 1-S4D.

Reviewer #4:Francisco et al. investigate the role of CTP and hydrolysis in the binding of ParB to parS sequence and non-specific DNA at the single-molecule level. Using optical tweezers, they show the specific binding of ParB to parS sites, and demonstrate that this process is enhanced by the presence of CTP or CTPγS. They find that lower density ParB proteins are also detected in distal non-specific DNA in the presence of parS, and that ParB spreading is restricted by protein roadblocks. Furthermore, using magnetic tweezers, they show that parS-containing DNA molecules are condensed by ParB at nanomolar protein concentration, which requires CTP binding but not hydrolysis. These finding show the significance of CTP-dependent ParB spreading and impact the understanding of the mechanism of DNA bridging and condensation by ParB networks.Based on these results, the authors propose a model for ParB-mediated DNA condensation, which requires one-dimensional ParB sliding along DNA from parS sites. Overall, the experiments were carefully done and thoroughly controlled. The manuscript provides critical insights that can be strengthened by addressing the following concerns:1. Did the authors observe the diffusion of isolated ParB foci along DNA? This will provide strong evidence for the proposed diffusion/sliding model.

This is an important point, also raised by the other reviewers. Please see our detailed answer to this query in the “essential revisions” section elaborated by the editor.

2. Based on the sliding clamp model, ParB spreading and diffusion result in DNA condensation by forming large DNA loops. Is it possible to show the dynamic spreading of ParB while keep the same numbers of ParB on DNA? For example, can the authors incubate ParB-containing DNA in channel 4 (ParB channel) at a certain time for the loading of ParB on parS sites, and then move it to the buffer channel without free ParB as well as with CTP or CTPγS, where the images are acquired at the long interval time to minimize the photobleaching. The fluorescent intensity of the ParB during the spreading process can be analyzed. If the intensity remains constant through spreading in the presence of CTPγS but significantly decrease in the presence of CTP, this data will strongly demonstrate the proposed spreading and CTP hydrolysis-dependent dissociation mechanism.

We thank the reviewer for these suggestions to prove spreading. However, we decided to follow an alternative strategy based on the direct imaging of QD-labelled ParB. As described above, this strategy worked well and we have directly visualized ParB diffusion from parS sites.

3. In Figure 2, the authors show the spreading of ParB can be blocked by EcoRI. Can the authors show that EcoRI is bound at the specificity positions? The spreading blockage by protein roadblocks showed in optical tweezers experiments potentially hints that the roadblocks may affect the DNA condensation. Can the authors apply the magnetic tweezers to show the affection of protein roadblocks to DNA condensation in vitro?

It is well established that EcoRI has extremely high affinity and specificity for its site (Terry et al., 1983) and so, since we do not have labelled EcoRI mutant, our experiments assume the sites are occupied. This is one reason we have used multiple sites in our experiments. Nevertheless, we have tested the effect of protein roadblocks in condensation in MT experiments. We found partial concentration consistent with the blocking of spreading of ParB from parS (Figure 7-figure supplement 1).

References

Breier AM, Grossman AD. 2007. Whole-genome analysis of the chromosome partitioning and sporulation protein Spo0J (ParB) reveals spreading and origin-distal sites on the *Bacillus subtilis* chromosome. *Mol Microbiol* 64:703–718. doi:10.1111/j.1365-2958.2007.05690.x

Graham TG, Wang X, Song D, Etson CM, van Oijen AM, Rudner DZ, Loparo JJ. 2014. ParB spreading requires DNA bridging. *Genes Dev* 28:1228–1238. doi:10.1101/gad.242206.114

Guilhas B, Walter J-C, Rech J, David G, Walliser NO, Palmeri J, Mathieu-Demaziere C, Parmeggiani A, Bouet J-Y, Le Gall A, Nollmann M. 2020. ATP-Driven Separation of Liquid Phase Condensates in Bacteria. *Mol Cell* 79:293-303.e4. doi:https://doi.org/10.1016/j.molcel.2020.06.034

Livny J, Yamaichi Y, Waldor MK. 2007. Distribution of centromere-like parS sites in bacteria: insights from comparative genomics. *J Bacteriol* 189:8693–8703. doi:10.1128/JB.01239-07

Terry BJ, Jack WE, Rubin RA, Modrich P. 1983. Thermodynamic parameters governing interaction of EcoRI endonuclease with specific and nonspecific DNA sequences. *J Biol Chem* 258:9820–9825.